# ENHANCING ADVERSARIAL DEFENSE BY $k$-WINNERS-TAKE-ALL

**Chang Xiao**    **Peilin Zhong**    **Changxi Zheng**
Columbia University
`{chang, peilin, cxz}@cs.columbia.edu`

## ABSTRACT

We propose a simple change to existing neural network structures for better defending against gradient-based adversarial attacks. Instead of using popular activation functions (such as ReLU), we advocate the use of $k$-Winners-Take-All ($k$-WTA) activation, a $C^0$ discontinuous function that purposely invalidates the neural network model's gradient at densely distributed input data points. The proposed $k$-WTA activation can be readily used in nearly all existing networks and training methods with no significant overhead. Our proposal is theoretically rationalized. We analyze why the discontinuities in $k$-WTA networks can largely prevent gradient-based search of adversarial examples and why they at the same time remain innocuous to the network training. This understanding is also empirically backed. We test $k$-WTA activation on various network structures optimized by a training method, be it adversarial training or not. In all cases, the robustness of $k$-WTA networks outperforms that of traditional networks under white-box attacks.

## 1 INTRODUCTION

In the tremendous success of deep learning techniques, there is a grain of salt. It has become well-known that deep neural networks can be easily fooled by *adversarial examples* (Szegedy et al., 2014). Those deliberately crafted input samples can mislead the networks to produce an output drastically different from what we expect. In many important applications, from face recognition authorization to autonomous cars, this vulnerability gives rise to serious security concerns (Barreno et al., 2010; 2006; Sharif et al., 2016; Thys et al., 2019).

Attacking the network is straightforward. Provided a labeled data item $(\boldsymbol{x}, y)$, the attacker finds a perturbation $\boldsymbol{x}'$ perceptually similar to $\boldsymbol{x}$ but misleading enough to cause the network to output a label different from $y$. By far, the most effective way of finding such a perturbation (or adversarial example) is by exploiting the gradient information of the network with respect to its input: the gradient indicates how to perturb $\boldsymbol{x}$ to trigger the maximal change of $y$.

The defense, however, is challenging. Recent studies showed that adversarial examples always exist if one tends to pursue a high classification accuracy—adversarial robustness seems at odds with the accuracy (Tsipras et al., 2018; Shafahi et al., 2019a; Su et al., 2018a; Weng et al., 2018; Zhang et al., 2019). This intrinsic difficulty of eliminating adversarial examples suggests an alternative path: *can we design a network whose adversarial examples are evasive rather than eliminated?* Indeed, along with this thought is a series of works using obfuscated gradients as a defense mechanism (Athalye et al., 2018). Those methods hide the network's gradient information by artificially discretizing the input (Buckman et al., 2018; Lin et al., 2019) or introducing certain randomness to the input (Xie et al., 2018a; Guo et al., 2018) or the network structure (Dhillon et al., 2018; Cohen et al., 2019) (see more discussion in Sec. 1.1). Yet, the hidden gradient in those methods can still be approximated, and as recently pointed out by Athalye et al. (2018), those methods remain vulnerable.

**Technical contribution I)**    Rather than obfuscating the network's gradient, we make the gradient *undefined*. This is achieved by a simple change to the standard neural network structure: we advocate the use of a $C^0$ *discontinuous* activation function, namely the $k$-Winners-Take-All ($k$-WTA) activation, to replace the popular activation functions such as rectified linear units (ReLU). This is the only change we propose to a deep neural network. All other components (such as BatchNorm, convolution, and pooling) as well as the training methods remain unaltered. With no significant overhead, $k$-WTA activation can be readily used in nearly all existing networks and training methods.

$k$-WTA activation takes as input the entire output of a layer, retains its $k$ largest values and deactivates all others to zero. As we will show in this paper, even an infinitesimal perturbation to the input may cause a complete change to the network neurons' activation pattern, thereby resulting in a large jump in the network's output. This means that, mathematically, if we use $f(\boldsymbol{x}; \boldsymbol{w})$ to denote a $k$-WTA network taking an input $\boldsymbol{x}$ and parameterized by weights $\boldsymbol{w}$, then the gradient $\nabla_{\boldsymbol{x}} f(\boldsymbol{x}; \boldsymbol{w})$ at certain $\boldsymbol{x}$ is undefined—$f(\boldsymbol{x}; \boldsymbol{w})$ is $C^0$ *discontinuous*.

**Technical contribution II)** More intriguing than the mere replacement of the activation function is *why $k$-WTA helps improve the adversarial robustness*. We offer our theoretical reasoning of its behavior from two perspectives. On the one hand, we show that the discontinuities of $f(\boldsymbol{x}; \boldsymbol{w})$ is *densely distributed* in the space of $\boldsymbol{x}$. Dense enough such that a tiny perturbation from any $\boldsymbol{x}$ almost always comes across some discontinuities, where the gradients are undefined and thus the attacker's search of adversarial examples becomes blinded (see Figure 1).

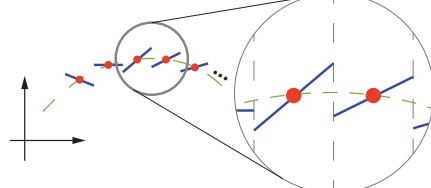

Figure 1: **1D illustration.** Fit a 1D function (green dotted curve) using a $k$-WTA model provided with a set of points (red). The resulting model is piecewise continuous (blue curve), and the discontinuities can be dense.

On the other hand, a paradox seemingly exists. The discontinuities in the activation function also renders $f(\boldsymbol{x}; \boldsymbol{w})$ discontinuous with respect to the network weights $\boldsymbol{w}$ (at certain $\boldsymbol{w}$ values). But training the network relies on the presumption that the gradient with respect to the weights is almost always available. Interestingly, we show that, under $k$-WTA activation, the discontinuities of $f(\boldsymbol{x}; \boldsymbol{w})$ is rather *sparse* in the space of $\boldsymbol{w}$, intuitively because the dimension of $\boldsymbol{w}$ (in parameter space) is much larger than the dimension of $\boldsymbol{x}$ (in data space). Thus, the network can be trained successfully.

**Summary of results.** We conducted extensive experiments on multiple datasets under different network architectures, including ResNet (He et al., 2016), DenseNet (Huang et al., 2017), and Wide ResNet (Zagoruyko & Komodakis, 2016), that are optimized by regular training as well as various adversarial training methods (Madry et al., 2017; Zhang et al., 2019; Shafahi et al., 2019b).

In all these setups, we compare the robustness performance of using the proposed $k$-WTA activation with commonly used ReLU activation under several white-box attacks, including PGD (Kurakin et al., 2016), Deepfool (Moosavi-Dezfooli et al., 2016), C&W (Carlini & Wagner, 2017), MIM (Dong et al., 2018), and others. In all tests, $k$-WTA networks outperform ReLU networks.

The use of $k$-WTA activation is motivated for defending against gradient-based adversarial attacks. Our experiments suggest that the robustness improvement gained by simply switching to $k$-WTA activation is universal, not tied to specific network architectures or training methods. To promote reproducible research, we will release our implementation of $k$-WTA networks, along with our experiment code, configuration files and pre-trained models[1].

## 1.1 RELATED WORK: OBFUSCATED GRADIENTS AS A DEFENSE MECHANISM

Before delving into $k$-WTA details, we review prior adversarial defense methods that share the same philosophy with our method and highlight our advantages. For a review of other attack and defense methods, we refer to Appendix A.

Methods aiming for concealing the gradient information from the attacker has been termed as *obfuscated gradients* (Athalye et al., 2018) or *gradient masking* (Papernot et al., 2017; Tramèr et al., 2017) techniques. One type of such methods is by exploiting randomness, either randomly transforming the input before feeding it to the network (Xie et al., 2018a; Guo et al., 2018) or introducing stochastic layers in the network (Dhillon et al., 2018). However, the gradient information in these methods can be estimated by taking the average over multiple trials (Athalye et al., 2018; 2017). As a result, these methods are vulnerable.

Another type of obfuscated gradient methods relies on the so-called *shattered gradient* (Athalye et al., 2018), which aims to make the network gradients nonexistent or incorrect to the attacker, by purposely discretizing the input (Buckman et al., 2018; Ma et al., 2018) or artificially raising numerical instability for gradient evaluation (Song et al., 2018; Samangouei et al., 2018). Unfortunately, these methods are also vulnerable. As shown by Athalye et al. (2018), they can be compromised by *backward pass*

---

[1]https://github.com/a554b554/kWTA-Activation

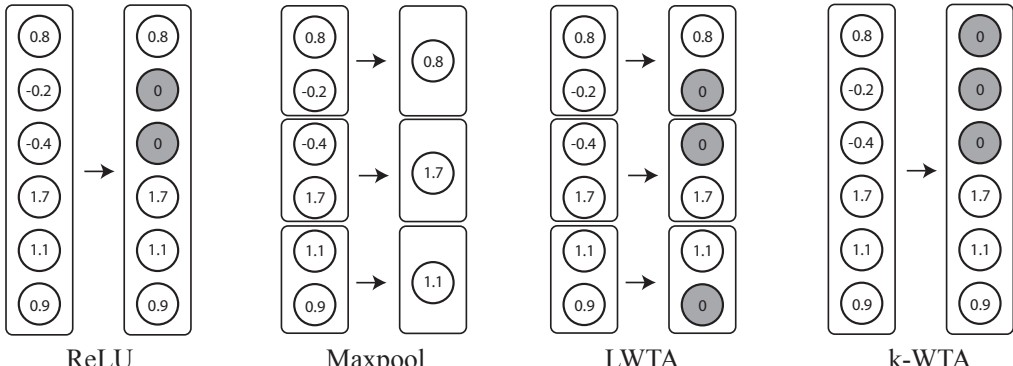

Figure 2: **Different activation functions. ReLU**: all neurons with negative activation values will be set to zero. **Max-pooling**: only the largest activation in each group is transmitted to the next layer, and this effectively downsample the output. **LWTA**: the largest activation in each group retains its value when entering the next layer, others are set to zero. $k$**-WTA**: the $k$ largest activations in the entire layer retain their values when entering the next layer, others are set to zero ($k = 3$ in this example). Note that the output is not downsampled through ReLU, LWTA and $k$-WTA.

*differentiable approximation* (BPDA). Suppose $f_i(\boldsymbol{x})$ is a non-differentiable component of a network expressed by $f = f_1 \circ f_2 \circ \cdots \circ f_n$. The gradient $\nabla_{\boldsymbol{x}} f$ can be estimated as long as one can find a smooth delegate $g$ that approximates well $f_i$ (i.e., $g(x) \approx f_i(x)$).

In stark contrast to all those methods, a slight change of the $k$-WTA activation pattern in an earlier layer of a network can cause a radical reorganization of activation patterns in later layers (as shown in Sec. 3). Thereby, $k$-WTA activation not just obfuscates the network's gradients but destroys them at certain input samples, introducing discontinuities densely distributed in the input data space. We are not aware of any possible smooth approximation of a $k$-WTA network to launch BPDA attacks.

## 2 $k$-WINNERS-TAKE-ALL ACTIVATION

The debut of the *Winner-Takes-All* (WTA) activation on the stage of neural networks dates back to 1980s, when Grossberg (1982) introduced shunting short-term memory equations in on-center off-surround networks and showed the ability to identify the largest of $N$ real numbers. Later, Majani et al. (1989) generalized the WTA network to identify the $K$ largest of $N$ real numbers, and they termed the network as the K-Winners-Take-All (KWTA) network. These early WTA-type activation functions output only boolean values, mainly motivated by the properties of biological neural circuits. In particular, Maass (2000a;b) has proved that any boolean function can be computed by a single KWTA unit. Yet, the boolean nature of these activation functions differs starkly from the modern activation functions, including the one we use.

### 2.1 DEEP NEURAL NETWORKS ACTIVATED BY $k$-WINNERS-TAKE-ALL

We propose to use $k$-Winners-Take-All ($k$-WTA) activation, a natural generalization of the boolean KWTA[2] (Majani et al., 1989). $k$-WTA retains the $k$ largest values of an $N \times 1$ input vector and sets all others to be zero before feeding the vector to the next network layer, namely,

$$\phi_k(\boldsymbol{y})_j = \begin{cases} y_j, & y_j \in \{k \text{ largest elements of } \boldsymbol{y}\}, \\ 0, & \text{Otherwise.} \end{cases} \quad (1)$$

Here $\phi_k : \mathbb{R}^N \to \mathbb{R}^N$ is the $k$-WTA function (parameterized by an integer $k$), $\boldsymbol{y} \in \mathbb{R}^N$ is the input to the activation, and $\phi_k(\boldsymbol{y})_j$ denote the $j$-the element of the output $\phi_k(\boldsymbol{y})$ (see the rightmost subfigure of Figure 2). Note that if $\boldsymbol{y}$ has multiple elements that are equally $k$-th largest, we break the tie by retaining the element with smaller indices until the $k$ slots are taken.

When using $k$-WTA activation, we need to choose $k$. Yet it makes no sense to fix $k$ throughout all layers of the neural network, because these layers often have different output dimensions; a small $k$ to one layer's dimension can be relatively large to the other. Instead of specifying $k$, we introduce

---

[2]In this paper, we use $k$-WTA to refer our activation function, while using KWTA to refer the original boolean version by Majani et al. (1989).

a parameter $\gamma \in (0, 1)$ called *sparsity ratio*. If a layer has an output dimension $N$, then its $k$-WTA activation has $k = \lfloor \gamma \cdot N \rfloor$. Even though the sparsity ratio can be set differently for different layers, in practice we found no clear gain from introducing such a variation. Therefore, we use a fixed $\gamma$—the only additional hyperparameter needed for the neural network.

In convolutional neural networks (CNN), the output of a layer is a $C \times H \times W$ tensor. $C$ denotes the number of output channels; $H$ and $W$ indicate the feature resolution. While there are multiple choices of applying $k$-WTA on the tensor—for example, one can apply $k$-WTA individually to each channel—empirically we found that the most effective (and conceptually the simplest) way is to treat the tensor as a long $C \cdot H \cdot W \times 1$ vector input to the $k$-WTA activation. Using $k$-WTA in this way is also backed by our theoretical understanding (see Sec. 3).

The runtime cost of computing a $k$-WTA activation is asymptotically $O(N)$, because finding $k$ largest values in a list is asymptotically equivalent to finding the $k$-th largest value, which has an $O(N)$ complexity (Cormen et al., 2009). This cost is comparable to ReLU's $O(N)$ cost on a $N$-length vector. Thus, replacing ReLU with $k$-WTA introduces no significant overhead.

**Remark: other WTA-type activations.**   Relevant to $k$-WTA is the *local Winner-Take-All* (LWTA) activation (Srivastava et al., 2013; 2014), which divides each layer's output values into local groups and applies WTA to each group individually. LWTA is similar to max-pooling (Riesenhuber & Poggio, 1999) for dividing the layer output and choosing group maximums. But unlike ReLU and max-pooling being $C^0$ continuous, LWTA and our $k$-WTA are both discontinuous with respect to the input. The differences among ReLU, max-pooling, LWTA, and $k$-WTA are illusrated in Figure 2.

LWTA is motivated toward preventing catastrophic forgetting (McCloskey & Cohen, 1989), whereas our use of $k$-WTA is for defending against adversarial threat. Both are discontinuous. But it remains unclear what the LWTA's discontinuity properties are and how its discontinuities affect the network training. Our theoretical analysis (Sec. 3), in contrast, sheds some light on these fundamental questions about $k$-WTA, rationalizing its ability for improving adversarial robustness. Indeed, our experiments confirm that $k$-WTA outperforms LWTA in terms of robustness (see Appendix D.3).

WTA-type activation, albeit originated decades ago and widely studied in computational neuro-science (Douglas et al., 1989; Douglas & Martin, 2004), remains elusive in modern neural networks. Perhaps this is because it has not demonstrated a considerable improvement to the network's standard test accuracy, though it can offer an accuracy comparable to ReLU (Srivastava et al., 2013). Our analysis and proposed use of $k$-WTA and its enabled improvement on adversarial defense may suggest a renaissance of studying WTA.

## 2.2   TRAINING $k$-WTA NETWORKS

$k$-WTA networks require no special treatment in training. Any optimization algorithm (such as stochastic gradient descent) for training ReLU networks can be directly used to train $k$-WTA networks.

Our experiments have found that when the sparsity ratio $\gamma$ is relatively small ($\leq 0.2$), the network training converges slowly. This is not a surprise. A smaller $\gamma$ activates fewer neurons, effectively reducing more of the layer width and in turn the network size, and the stripped "subnetwork" is much less expressive (Srivastava et al., 2013). Since different training examples activate different subnetworks, collectively they make the training harder.

Nevertheless, we prefer a smaller $\gamma$. As we will discuss in the next section, a smaller $\gamma$ usually leads to better robustness against finding adversarial examples. Therefore, to ease the training (when $\gamma$ is small), we propose to use an iterative fine-tuning approach. Suppose the target sparsity ratio is $\gamma_1$. We first train the network with a larger sparsity ratio $\gamma_0$ using the standard training process. Then, we iteratively fine tune the network. In each iteration, we reduce its sparsity ratio by a small $\delta$ and train the network for two epochs. The iteration repeats until $\gamma_0$ is reduced to $\gamma_1$.

This incremental process introduces little training overhead, because the cost of each fine tuning is negligible in comparison to training from scratch toward $\gamma_0$. We also note that this process is optional. In practice we use it only when $\gamma < 0.2$. We show more experiments on the efficacy of the incremental training in Appendix D.2.

## 3   UNDERSTANDING $k$-WTA DISCONTINUITY

We now present our theoretical understanding of $k$-WTA's discontinuity behavior in the context of deep neural networks, revealing some implication toward the network's adversarial robustness.

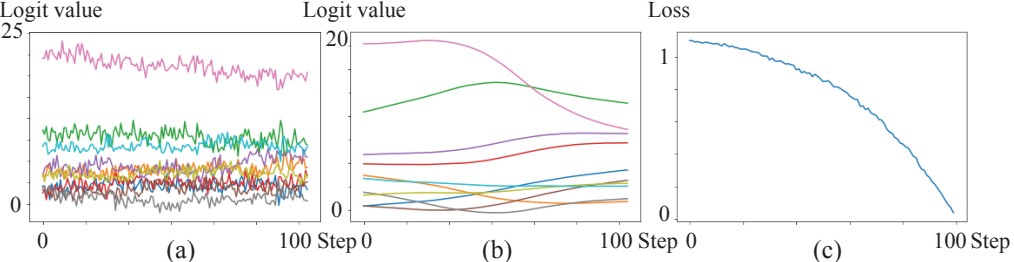

Figure 3: **(a, b)** We plot the change of 10 logits values when conducting untargeted PGD attack with 100 iterations. X-axis indicates the perturbation size $\epsilon$ and Y-axis indicates the 10 color-coded logits values. **(a)** When we apply PGD attack on $k$-WTA ResNet18, the strong discontinuities w.r.t. to input invalidate gradient estimation, effectively defending well against the attack. **(b)** In contrast, for a ReLU ResNet18, PGD attack can easily find adversarial examples due to the model's smooth change w.r.t. input. **(c)** In the process of training $k$-WTA ResNet18, the loss change w.r.t. model weights is largely smooth. Thus, the training is not harmed by $k$-WTA's discontinuities.

**Activation pattern.** To understand $k$-WTA's discontinuity, consider one layer outputting values $\boldsymbol{x}$, passed through a $k$-WTA activation, and followed by the next layer whose linear weight matrix is $\mathsf{W}$ (see adjacent figure). Then, the value fed into the next activation can be expressed as $\mathsf{W}\phi_k(\boldsymbol{x})$, where $\phi_k(\cdot)$ is the $k$-WTA function defined in (1). Suppose the vector $\boldsymbol{x}$ has a length $l$. We define the $k$-WTA's *activation pattern* under the input $\boldsymbol{x}$ as

$$\mathcal{A}(\boldsymbol{x}) \coloneqq \{i \in [l] \mid x_i \text{ is one of the } k \text{ largest values in } \boldsymbol{x}\} \subseteq [l]. \tag{2}$$

Here (and throughout this paper), we use $[l]$ to denote the integer set $\{1, 2, ..., l\}$.

**Discontinuity.** The activation pattern $\mathcal{A}(\boldsymbol{x})$ is a key notion for analyzing $k$-WTA's discontinuity behavior. Even an infinitesimal perturbation of $\boldsymbol{x}$ may change $\mathcal{A}(\boldsymbol{x})$: some element $i$ is removed from $\mathcal{A}(\boldsymbol{x})$ while another element $j$ is added in. Corresponding to this change, in the evaluation of $\mathsf{W}\phi_k(\boldsymbol{x})$, the contribution of $\mathsf{W}$'s column vector $\mathsf{W}_i$ vanishes while another column $\mathsf{W}_j$ suddenly takes effect. It is this abrupt change that renders the result of $\mathsf{W}\phi_k(\boldsymbol{x})$ $C^0$ discontinuous.

Such a discontinuity jump can be arbitrarily large, because the column vectors $\mathsf{W}_i$ and $\mathsf{W}_j$ can be of any difference. Once $\mathsf{W}$ is determined, the discontinuity jump then depends on the value of $x_i$ and $x_j$. As explained in Appendix B, when the discontinuity occurs, $x_i$ and $x_j$ have about the same value, depending on the choice of the sparsity ratio $\gamma$ (recall Sec. 2.1)—the smaller the $\gamma$ is, the larger the jump will be. This relationship suggests that a smaller $\gamma$ will make the search of adversarial examples harder. Indeed, this is confirmed through our experiments (see Appendix D.6).

**Piecewise linearity.** Now, consider an $n$-layer $k$-WTA network, which can be expressed as $f(\boldsymbol{x}) = \mathsf{W}^{(1)} \cdot \phi_k(\mathsf{W}^{(2)} \cdot \phi_k(\cdots \phi_k(\mathsf{W}^{(n)}\boldsymbol{x} + \boldsymbol{b}^{(n)})) + \boldsymbol{b}^{(2)}) + \boldsymbol{b}^{(1)}$, where $\mathsf{W}^{(i)}$ and $\boldsymbol{b}^{(i)}$ are the $i$-th layer's weight matrix and bias vector, respectively. If the activation patterns of all layers are fixed, then $f(\boldsymbol{x})$ is a linear function. When the activation pattern changes, $f(\boldsymbol{x})$ switches from one linear function to another linear function. Over the entire space of $\boldsymbol{x}$, $f(\boldsymbol{x})$ is *piecewise* linear. The specific activation patterns of all layers define a specific linear piece of the function, or a *linear region* (following the notion introduced by Montufar et al. (2014)). Conventional ReLU (or hard tanh) networks also represent piecewise linear functions and their linear regions are joined together at their boundaries, whereas in $k$-WTA networks the linear regions are disconnected (see Figure 1).

**Linear region density.** Next, we gain some insight on the distribution of those linear regions. This is of our interest because if the linear regions are densely distributed, a small $\Delta\boldsymbol{x}$ perturbation from any data point $\boldsymbol{x}$ will likely cross the boundary of the linear region where $\boldsymbol{x}$ locates. Whenever boundary crossing occurs, the gradient becomes undefined (see Figure 3-a).

For the purpose of analysis, consider an input $\boldsymbol{x}$ passing through a layer followed by a $k$-WTA activation (see adjacent figure). The output from the activation is $\phi_k(\mathsf{W}\boldsymbol{x} + \boldsymbol{b})$. We would like to understand, when $\boldsymbol{x}$ is changed into $\boldsymbol{x}'$, how likely the activation pattern of $\phi_k$ will change. First, notice that if $\boldsymbol{x}'$ and $\boldsymbol{x}$ satisfy $\boldsymbol{x}' = c \cdot \boldsymbol{x}$ with some $c > 0$, the activation pattern remains unchanged. Therefore, we introduce a notation $\mathsf{d}(\boldsymbol{x}, \boldsymbol{x}')$ that measures the "perpendicular" distance between $\boldsymbol{x}$ and $\boldsymbol{x}'$, one

that satisfies $\boldsymbol{x}' = c \cdot (\boldsymbol{x} + \mathsf{d}(\boldsymbol{x}, \boldsymbol{x}')\boldsymbol{x}_\perp)$ for some scalar $c$, where $\boldsymbol{x}_\perp$ is a unit vector perpendicular to $\boldsymbol{x}$ and on the plane spanned by $\boldsymbol{x}$ and $\boldsymbol{x}'$. With this notion, and if the elements in $\mathsf{W}$ is initialized by sampling from $\mathcal{N}(0, \frac{1}{l})$ and $\boldsymbol{b}$ is initialized as zero, we find the following property:

**Theorem 1** (Dense discontinuities). *Given any input $\boldsymbol{x} \in \mathbb{R}^m$ and some $\beta$, and $\forall \boldsymbol{x}' \in \mathbb{R}^m$ such that $\frac{\mathsf{d}^2(\boldsymbol{x}, \boldsymbol{x}')}{\|\boldsymbol{x}\|_2^2} \geq \beta$, if the following condition*

$$l \geq \Omega\left(\left(\frac{m}{\gamma} \cdot \frac{1}{\beta}\right) \cdot \log\left(\frac{m}{\gamma} \cdot \frac{1}{\beta}\right)\right)$$

*is satisfied, then with a probability at least $1 - \cdot 2^{-m}$, we have $\mathcal{A}(\mathsf{W}\boldsymbol{x} + \boldsymbol{b}) \neq \mathcal{A}(\mathsf{W}\boldsymbol{x}' + \boldsymbol{b})$.*

Here $l$ is the width of the layer, and $\gamma$ is again the sparsity ratio in $k$-WTA. This theorem informs us that the larger the layer width $l$ is, the smaller $\beta$—and thus the smaller perpendicular perturbation distance $\mathsf{d}(\boldsymbol{x}, \boldsymbol{x}')$—is needed to trigger a change of the activation pattern, that is, as the layer width increases, the piecewise linear regions become finer (see Appendix C for proof and more discussion). This property also echos a similar trend in ReLU networks, as pointed out by Raghu et al. (2017).

**Why is the $k$-WTA network trainable?** While $k$-WTA networks are highly discontinuous as revealed by Theorem 1 and our experiments (Figure 3-a), in practice we experience no difficulty on training these networks. Our next theorem sheds some light on the reason behind the training success.

**Theorem 2.** *Consider $N$ data points $\boldsymbol{x}_1, \boldsymbol{x}_2, \cdots, \boldsymbol{x}_N \in \mathbb{R}^m$. Suppose $\forall i \neq j, \frac{\boldsymbol{x}_i}{\|\boldsymbol{x}_i\|_2} \neq \frac{\boldsymbol{x}_j}{\|\boldsymbol{x}_j\|_2}$. If $l$ is sufficiently large, then with a high probability, we have $\forall i \neq j, \mathcal{A}(\mathsf{W}\boldsymbol{x}_i + \boldsymbol{b}) \cap \mathcal{A}(\mathsf{W}\boldsymbol{x}_j + \boldsymbol{b}) = \varnothing$.*

This theorem is more formally stated in Theorem 10 in Appendix C together with a proof there. Intuitively speaking, it states that if the network is sufficiently wide, then for any $i \neq j$, activation pattern of input data $\boldsymbol{x}_i$ is almost separated from that of $\boldsymbol{x}_j$. Thus, the weights for predicting $\boldsymbol{x}_i$'s and $\boldsymbol{x}_j$'s labels can be optimized almost independently, without changing their individual activation patterns. In practice, the activation patterns of $\boldsymbol{x}_i$ and $\boldsymbol{x}_j$ are not fully separated but weakly correlated. During the optimization, the activation pattern of a data point $\boldsymbol{x}_i$ may change, but the chance is relatively low—a similar behavior has also been found in ReLU networks (Li & Liang, 2018; Du et al., 2018; Allen-Zhu et al., 2019a;b; Song & Yang, 2019).

Further, notice that the training loss takes a summation over all training data points. This means a weight update would change only a small set of activation patterns (since the chance of having the pattern changed is low); the discontinuous change on the loss value, after taking the summation, will be negligible (see Figure 3-c). Thus, the discontinuities in $k$-WTA is not harmful to network training.

## 4 EXPERIMENTAL RESULTS

We evaluate the robustness of $k$-WTA networks under adversarial attacks. Our evaluation considers multiple training methods on different network architectures (see details below). When reporting statistics, we use $A_{rob}$ to indicate the *worst-case* robustness accuracy on test data under all adversarial attacks we evaluated, and use $A_{std}$ to indicate the accuracy on the clean test data. We use $k$-WTA-$\gamma$ to represent $k$-WTA activation with sparsity ratio $\gamma$.

### 4.1 ROBUSTNESS UNDER WHITE-BOX ATTACKS

The rationale behind $k$-WTA activation is to destroy network gradients—information needed in white-box attacks. We therefore evaluate $k$-WTA networks under multiple recently proposed white-box attack methods, including *Projected Gradient Descent* (PGD) (Madry et al., 2017), *Deepfool* (Moosavi-Dezfooli et al., 2016), C&W attack (Carlini & Wagner, 2017), and *Momentum Iterative Method* (MIM) (Dong et al., 2018). Since $k$-WTA activation can be used in almost any training method, be it adversarial training or not, we also consider multiple training methods, including natural (non-adversarial) training, adversarial training (AT) (Madry et al., 2017), TRADES (Zhang et al., 2019) and free adversarial training (FAT) (Shafahi et al., 2019b).

In addition, we evaluate the robustness under transfer-based Black-box (BB) attacks (Papernot et al., 2017). The black-box threat model requires no knowledge about network architecture and parameters. Thus, we use a pre-trained VGG19 network (Simonyan & Zisserman, 2014) as the source model to generate adversarial examples using PGD. As demonstrated by Su et al. (2018b), VGG networks have the strongest transferability among different architectures.

Table 1: Adversarial robustness on CIFAR-10 and SVHN datasets. $A_{rob}$ in the last column denotes the empirical worst-case robustness among different attacks (columns) for each network optimized by different training methods (row). The **bold** numbers indicate the best $A_{rob}$ robustness achieved on ReLU and $k$-WTA networks by each training method.

**CIFAR-10**

| Training | Activation | $A_{std}$ | PGD | C&W | Deepfool | MIM | BB | $A_{rob}$ |
|---|---|---|---|---|---|---|---|---|
| | ReLU | 92.9% | 0.0% | 0.0% | 1.5% | 0.0% | 18.9% | 0.0% |
| Natural | $k$-WTA-0.1 | 89.3% | 13.3% | 27.9% | 55.6% | 13.1% | 62.6% | **13.1%** |
| | $k$-WTA-0.2 | 91.7% | 4.2% | 6.2% | 47.8% | 3.9% | 66.8% | 4.2% |
| | ReLU | 83.5% | 46.3% | 43.6% | 46.8% | 45.9% | 71.0% | 43.6% |
| AT | $k$-WTA-0.1 | 78.9% | 51.4% | 64.4% | 70.4% | 50.7% | 73.4% | **50.7%** |
| | $k$-WTA-0.2 | 81.4% | 48.4% | 52.7% | 66.1% | 47.4% | 73.5% | 47.4% |
| | ReLU | 79.7% | 49.8% | 52.3% | 57.6% | 49.9% | 70.6% | 49.8% |
| TRADES | $k$-WTA-0.1 | 76.6% | 55.0% | 62.2% | 66.0% | 57.5% | 72.3% | **55.0%** |
| | $k$-WTA-0.2 | 80.4% | 51.5% | 57.7% | 63.9% | 53.4% | 74.7% | 51.5% |
| | ReLU | 82.6% | 42.7% | 44.4% | 49.7% | 41.6% | 73.4% | 41.6% |
| FAT | $k$-WTA-0.1 | 78.4% | 51.7% | 66.3% | 72.4% | 49.1% | 72.3% | **49.1%** |
| | $k$-WTA-0.2 | 82.8% | 48.4% | 60.5% | 67.2% | 46.7% | 76.8% | 46.7% |

**SVHN**

| Training | Activation | $A_{std}$ | PGD | C&W | Deepfool | MIM | BB | $A_{rob}$ |
|---|---|---|---|---|---|---|---|---|
| | ReLU | 95.1% | 0.0% | 0.0% | 2.5% | 0.0% | 14.7% | 0.0% |
| Natural | $k$-WTA-0.1 | 92.6% | 10.2% | 19.5% | 88.7% | 11.6% | 51.4% | **10.2%** |
| | $k$-WTA-0.2 | 93.8% | 4.3% | 8.0% | 86.8% | 8.3% | 56.7% | 4.3% |
| | ReLU | 84.2% | 44.5% | 42.7% | 70.3% | 48.4% | 77.7% | 42.7% |
| AT | $k$-WTA-0.1 | 79.9% | 62.2% | 65.7% | 71.5% | 56.9% | 76.1% | **56.9%** |
| | $k$-WTA-0.2 | 82.4% | 53.2% | 63.6% | 77.4% | 52.3% | 74.2% | 52.3% |
| | ReLU | 84.7% | 47.4% | 51.6% | 76.9% | 49.6% | 76.5% | 47.4% |
| TRADES | $k$-WTA-0.1 | 81.6% | 61.3% | 77.4% | 79.4% | 58.3% | 78.1% | **58.3%** |
| | $k$-WTA-0.2 | 85.4% | 56.7% | 59.2% | 71.6% | 54.5% | 79.3% | 54.5% |
| | ReLU | 85.9% | 40.8% | 46.2% | 76.1% | 39.9% | 76.9% | 40.8% |
| FAT | $k$-WTA-0.1 | 85.5% | 57.7% | 70.0% | 77.0% | 62.8% | 75.6% | **57.7%** |
| | $k$-WTA-0.2 | 86.8% | 54.3% | 64.3% | 74.7% | 55.2% | 74.4% | 54.3% |

In each setup, we compare the robust accuracy of $k$-WTA networks with standard ReLU networks on three datasets, CIFAR-10, SVHN, and MNIST. Results on the former two are reported in Table 1, while the latter is reported in Appendix D.4. We use ResNet-18 for CIFAR-10 and SVHN. The perturbation range is 0.031 (CIFAR-10) and 0.047 (SVHN) for pixels ranging in $[0, 1]$. More detailed training and attacking settings are reported in Appendix D.1.

**The main takeaway** from these experiments (in Table 1) is that $k$-WTA is able to universally improve the white-box robustness, regardless of the training methods. The $k$-WTA robustness under black-box attacks is not always significantly better than ReLU networks. But black-box attacks, due to the lack of network information, are generally much harder than white-box attacks. In this sense, white-box attacks make the networks more vulnerable, and $k$-WTA is able to improve a network's worst-case robustness. This improvement is not tied to any specific training method, achieved with no significant overhead, just by a simple replacement of ReLU with $k$-WTA.

Athalye et al. (2018) showed that gradient-based defenses may render the network more vulnerable under black-box attacks than under gradient-based white-box attacks. However, we have not observed this behavior in $k$-WTA networks. Even under the strongest black-box attack, i.e., by generating adversarial examples from an independently trained copy of the target network, gradient-based attacks are still stronger (with higher successful rate) than black-box attacks (see Appendix D.3).

Additional experiments include: **1)** tests under transfer attacks across two independently trained $k$-WTA networks and across $k$-WTA and ReLU networks, **2)** evaluation of $k$-WTA performance on different network architectures, and **3)** comparison of $k$-WTA performance with the LWTA (recall Sec. 2.1) performance. See Appendix D.3 for details.

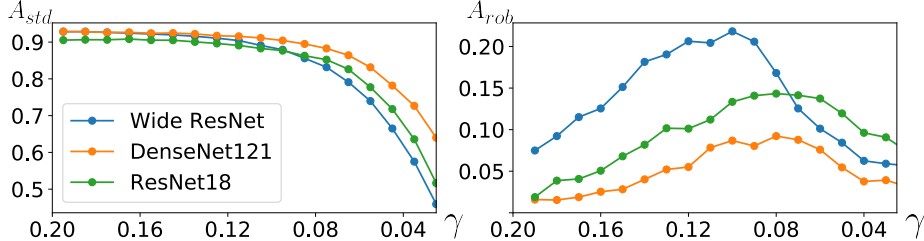

Figure 4: Robustness changing w.r.t. $\gamma$ on CIFAR. When $\gamma$ decreases, the standard test accuracy (left) starts to drop after a certain point. The robust accuracy (right) first increases then decreases.

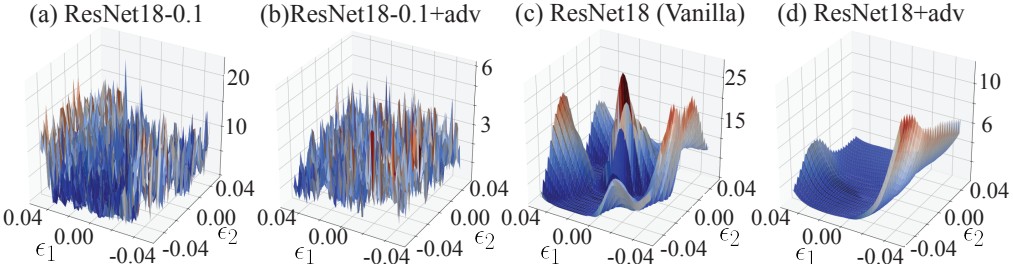

Figure 5: Gradient-based attack's loss landscapes in $k$-WTA **(a, b)** and conventional ReLU models **(c, d)**. (a,b) $k$-WTA Models have much more non-convex and non-smooth landscapes. Also, the model optimized by adversarial training (b) has a lower absolute value of loss.

## 4.2 VARYING SPARSITY RATIO $\gamma$ AND MODEL ARCHITECTURE.

We further evaluate our method on various network architectures with different sparsity ratios $\gamma$. Figure 4 shows the standard test accuracies and robust accuracies against PGD attacks while $\gamma$ decreases. To test on different network architectures, we apply $k$-WTA to ResNet18, DenseNet121 and Wide ResNet (WRN-22-12). In each case, starting from $\gamma = 0.2$, we decrease $\gamma$ using incremental fine-tuning. We then evaluate the robust accuracy on CIFAR dataset, taking 20-iteration PGD attacks with a perturbation range $\epsilon = 0.31$ for pixels ranging in $[0, 1]$.

We find that when $\gamma$ is larger than $\sim 0.1$, reducing $\gamma$ has little effect on the standard accuracy, but increases the robust accuracy. When $\gamma$ is smaller than $\sim 0.1$, reducing $\gamma$ drastically lowers both the standard and robust accuracies. The peaks in the $A_{rob}$ curves (Figure 4-right) are consistent with our theoretical understanding: Theorem 1 suggests that when $l$ is fixed, a smaller $\gamma$ tends to sparsify the linear region boundaries, exposing more gradients to the attacker. Meanwhile, as also discussed in Sec. 3, a smaller $\gamma$ leads to a larger discontinuity jump and thus tends to improve the robustness.

## 4.3 LOSS LANDSCAPE IN GRADIENT-BASED ATTACKS

We now empirically unveil why $k$-WTA is able to improve the network's robustness (in addition to our theoretical analysis in Sec. 3). Here we visualize the attacker's loss landscape in gradient-based attacks in order to reveal the landscape change caused by $k$-WTA. Similar to the analysis in Tramèr et al. (2017), we plot the attack loss of a model with respect to its input on points $\boldsymbol{x}' = \boldsymbol{x} + \epsilon_1 \boldsymbol{g}_1 + \epsilon_2 \boldsymbol{g}_2$, where $\boldsymbol{x}$ is a test sample from CIFAR test set, $\boldsymbol{g}_1$ is the direction of the loss gradient with respect to the input, $\boldsymbol{g}_2$ is another random direction, $\epsilon_1$ and $\epsilon_2$ sweep in the range of $[-0.04, 0.04]$, with 50 samples each. This results in a 3D landscape plot with 2500 data points (Figure 5).

As shown in Figure 5, $k$-WTA models (with $\gamma = 0.1$) have a highly non-convex and non-smooth loss landscape. Thus, the estimated gradient is hardly useful for adversarial searches. This explains why $k$-WTA models can effectively resist gradient-based attacks. In contrast, ReLU models have a much smoother loss surface, from which adversarial examples can be easily found using gradient descent.

Inspecting the range of loss values in Figure 5, we find that adversarial training tends to compress the loss landscape's dynamic range in both the gradient direction and the other random direction, making the dynamic range smaller than that of the models without adversarial training. This phenomenon has been observed in ReLU networks (Madry et al., 2017; Tramèr et al., 2017). Interestingly, $k$-WTA models manifest a similar behavior (Figure 5-a,b). Moreover, we find that in $k$-WTA models a larger $\gamma$ leads to a smoother loss surface than a smaller $\gamma$ (see Appendix D.6 for more details).

## 5 CONCLUSION

This paper proposes to replace widely used activation functions with the $k$-WTA activation for improving the neural network's robustness against adversarial attacks. This is the only change we advocate. The underlying idea is to embrace the discontinuities introduced by $k$-WTA functions to make the search for adversarial examples more challenging. Our method comes almost for free, harmless to network training, and readily useful in the current paradigm of neural networks.

**Acknowledgments.** This work was supported in part by the National Science Foundation (CAREER-1453101, 1816041, 1910839, 1703925, 1421161, 1714818, 1617955, 1740833), Simons Foundation (#491119 to Alexandr Andoni), Google Research Award, a Google PhD Fellowship, a Snap Research Fellowship, a Columbia SEAS CKGSB Fellowship, and SoftBank Group.

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

# Supplementary Document
# Enhancing Adversarial Defense by $k$-Winners-Take-All

## A    OTHER RELATED WORK

In this section, we briefly review the key ideas of attacking neural network models and existing defense methods based on adversarial training.

**Attack methods.**    Recent years have seen adversarial attack studied extensively. The proposed attack methods fall under two general categories, *white-box* and *black-box* attacks.

The white-box threat model assumes that the attacker knows the model's structure and parameters fully. This means that the attacker can exploit the model's gradient (with respect to the input) to find adversarial examples. A baseline of such attacks is the *Fast Gradient Sign Method* (FGSM) (Goodfellow et al., 2014), which constructs the adversarial example $\boldsymbol{x}'$ of a given labeled data $(\boldsymbol{x}, y)$ using a gradient-based rule:

$$\boldsymbol{x}' = \boldsymbol{x} + \epsilon \text{sign}(\nabla_{\boldsymbol{x}} L(f(\boldsymbol{x}), y)), \tag{3}$$

where $f(x)$ denotes the neural network model's output, $L(\cdot)$ is the loss function provided $f(x)$ and input label $y$, and $\epsilon$ is the perturbation range for the allowed adversarial example.

Extending FGSM, *Projected Gradient Descent* (PGD) (Kurakin et al., 2016) utilizes local first-order gradient of the network in a multi-step fashion, and is considered the "strongest" first-order adversary (Madry et al., 2017). In each step of PGD, the adversarial example is updated by a FGSM rule, namely,

$$\boldsymbol{x}'_{n+1} = \Pi_{\boldsymbol{x}' \in \Delta_\epsilon} \boldsymbol{x}'_n + \epsilon \text{sign}(\nabla_{\boldsymbol{x}} L(f(\boldsymbol{x}'_n), y)), \tag{4}$$

where $\boldsymbol{x}'_n$ is the adversarial examples after $n$ steps and $\Pi_{\boldsymbol{x} \in \Delta_\epsilon}(\boldsymbol{x}'_n)$ projects $\boldsymbol{x}'_n$ back into an allowed perturbation range $\Delta_\epsilon$ (such as an $\epsilon$ ball of $\boldsymbol{x}$ under certain distance measure). Other attacks include Deepfool (Moosavi-Dezfooli et al., 2016), C&W (Carlini & Wagner, 2017) and momentum-based attack (Dong et al., 2018). Those methods are all using first-order gradient information to construct adversarial samples.

The black-box threat model is a strict subset of the white-box threat model. It assumes that the attacker has no information about the model's architecture or parameters. Some black-box attack model allows the attacker to query the victim neural network to gather (or reverse-engineer) information. By far the most successful black-box attack is transfer attack (Papernot et al., 2017; Tramèr et al., 2017). The idea is to first construct adversarial examples on an adversarially trained network and then attack the black-box network model use these samples. There also exist some gradient-free black-box attack methods, such as boundary attack (Brendel et al., 2017; Chen & Jordan, 2019), one-pixel attack (Su et al., 2019) and local search attack (Narodytska & Kasiviswanathan, 2016). Those methods rely on repeatedly evaluating the model and are not as effective as gradient-based white-box attacks.

**Adversarial training.**    Adversarial training (Goodfellow et al., 2014; Madry et al., 2017; Kurakin et al., 2016; Huang et al., 2015) is by far the most successful method against adversarial attacks. It trains the network model with adversarial images generated during the training time. Madry et al. (2017) showed that adversarial training in essence solves the following min-max optimization problem:

$$\min_{f} \mathbb{E}\{ \max_{\boldsymbol{x}' \in \Delta_\epsilon} L(f(\boldsymbol{x}'), y) \}, \tag{5}$$

where $\Delta_\epsilon$ is the set of allowed perturbations of training samples, and $y$ denotes the true label of each training sample. Recent works that achieve state-of-the-art adversarial robustness rely on adversarial training  (Zhang et al., 2019; Xie et al., 2018b). However, adversarial training is notoriously slow because it requires finding adversarial samples on-the-fly at each training epoch. Its prohibitive cost makes adversarial training difficult to scale to large datasets such as ImageNet (Deng et al., 2009) unless enormous computation resources are available. Recently, Shafahi et al. (2019b) revise the adversarial training algorithm to make it has similar training time as regular training, while keep the standard and robust accuracy comparable to standard adversarial training.

**Regularization.** Another type of defense is based on regularizing the neural network, and many works of this type are combined with adversarial training. For example, *feature denoising* (Xie et al., 2018b) adds several denoise blocks to the network structure and trains the network with adversarial training. Zhang et al. (Zhang et al., 2019) explicitly added a regularization term to balance the trade-off between standard accuracy and robustness, obtaining state-of-the-art robust accuracy on CIFAR.

Some other regularization-based methods require no adversarial training. For example, Ross & Doshi-Velez (2017) proposed to regularize the gradient of the model with respect to its input; Zheng et al. (2016) generated adversarial samples by adding random Gaussian noise to input data. However, these methods are shown to be brittle under stronger iterative gradient-based attacks such as PGD (Zhang et al., 2019). In contrast, as demonstrated in our experiments, our method without using adversarial training is able to greatly improve robustness under PGD and other attacks.

## B  DISCONTINUITY JUMP OF $\mathsf{W}\phi_k(x)$

Consider a gradual and smooth change of the vector $\boldsymbol{x}$. For the ease of illustration, let us assume all the values in $\boldsymbol{x}$ are distinct. Because every element in $\boldsymbol{x}$ changes smoothly, when the activation pattern $\mathcal{A}(\boldsymbol{x})$ changes, the $k$-th largest and $k+1$-th largest value in $\boldsymbol{x}$ must swap: the previously $k$-th largest value is removed from the activation pattern, while the previously $k+1$-th largest value is added in the activation pattern. Let $i$ and $j$ denote the indices of the two values, that is, $x_i$ is previously the $k$-th largest and $x_j$ is previously the $k+1$-th largest. When this swap happens, $x_i$ and $x_j$ must be infinitesimally close to each other, and we use $x^*$ to indicate their common value.

This swap affects the computation of $\mathsf{W}\phi_k(\boldsymbol{x})$. Before the swap happens, $x_i$ is in the activation pattern but $x_j$ is not, therefore $\mathsf{W}_i$ takes effect but $\mathsf{W}_j$ does not. After the swap, $\mathsf{W}_j$ takes effect while $\mathsf{W}_j$ is suppressed. Therefore, the discontinuity jump due to this swap is $(\mathsf{W}_j - \mathsf{W}_i)x^*$.

When $\mathsf{W}$ is determined, the magnitude of the jump depends on $x^*$. Recall that $x^*$ is the $k$-th largest value in $\boldsymbol{x}$ when the swap happens. Thus, it depends on $k$ and in turn the sparsity ratio $\gamma$: the smaller the $\gamma$ is, the smaller $k$ is effectively used (for a fixed vector length). As a result, the $k$-th largest value becomes larger—when $k = 1$, the largest value of $\boldsymbol{x}$ is used as $x^*$.

## C  THEORETICAL PROOFS

In this section, we will prove Theorem 1 and Theorem 2. The formal version of the two theorems are Theorem 9 and Theorem 10 respectively.

**Notation.** We use $[n]$ to denote the set $\{1, 2, \cdots, n\}$. We use $\mathbf{1}(\mathcal{E})$ to indicate an indicator variable. If the event $\mathcal{E}$ happens, the value of $\mathbf{1}(\mathcal{E})$ is 1. Otherwise the value of $\mathbf{1}(\mathcal{E})$ is 0. For a weight matrix $W$, we use $W_i$ to denote the $i$-th row of $W$. For a bias vector $b$, we use $b_i$ to denote the $i$-th entry of $b$.

In this section, we show some behaviors of the $k$-WTA activation function. Recall that an $n$-layer neural network $f(x)$ with $k$-WTA activation function can be seen as the following:

$$f(x) = W^{(1)} \cdot \phi_k(W^{(2)} \cdot \phi_k(\cdots \phi_k(W^{(n)}x + b^{(n)})) + b^{(2)}) + b^{(1)}$$

where $W^{(i)}$ is the weight matrix, $b^{(i)}$ is the bias vector of the $i$-th layer, and $\phi(\cdot)$ is the $k$-WTA activation function, i.e., for an arbitrary vector $y$, $\phi_k(y)$ is defined as the following:

$$\phi_k(y)_j = \begin{cases} y_j, & \text{if } y_j \text{ is one of the top-}k \text{ largest values,} \\ 0, & \text{otherwise.} \end{cases}$$

For simplicity of the notation, if $k$ is clear in the context, we will just use $\phi(y)$ for short. Notice that if there is a tie in the above definition, we assume the entry with smaller index has larger value. For a vector $y \in \mathbb{R}^l$, we define the activation pattern $\mathcal{A}(y) \subseteq [l]$ as

$$\mathcal{A}(y) = \{i \in [l] \mid y_i \text{ is one of the top-}k \text{ largest values}\}.$$

Notice that if the activation pattern $\mathcal{A}(y)$ is different from $\mathcal{A}(y')$, then $W \cdot \phi(y)$ and $W \cdot \phi(y')$ will be in different linear region. Actually, $W \cdot \phi(y)$ may even represent a discontinuous function. In the next section, we will show that when the network is much wider, the function may be more discontinuous with respect to the input.

### C.1 DISCONTINUITY WITH RESPECT TO THE INPUT

We only consider the activation pattern of the output of one layer. We consider the behavior of the network after the initialization of the weight matrix and the bias vector. By initialization, the entries of the weight matrix $W$ are i.i.d. random Gaussian variables, and the bias vector is zero. We can show that if the weight matrix is sufficiently wide, then for any vector $x$, with high probability, for all vector $x'$ satisfying that the "perpendicular" distance between $x$ and $x'$ is larger than a small threshold, the activation patterns of $Wx$ and $Wx'$ are different.

Notice that the scaling of $W$ does not change the activation pattern of $Wx$ for any $x$, we can thus assume that each entry of $W$ is a random variable with standard Gaussian distribution $N(0, 1)$.

Before we prove Theorem 9, let us prove several useful lemmas. The following several lemmas does not depend on the randomness of the weight matrix.

**Lemma 1** (Inputs with the same activation pattern form a convex set). *Given an arbitrary weight matrix $W \in \mathbb{R}^{l \times m}$ and an arbitrary bias vector $b \in \mathbb{R}^l$, for any $x \in \mathbb{R}^m$, the set of all the vectors $x' \in \mathbb{R}^m$ satisfying $\mathcal{A}(Wx' + b) = \mathcal{A}(Wx + b)$ is convex, i.e., the set*

$$\{x' \in \mathbb{R}^m \mid \mathcal{A}(Wx + b) = \mathcal{A}(Wx' + b)\}$$

*is convex.*

*Proof.* If $\mathcal{A}(Wx' + b) = \mathcal{A}(Wx + b)$, then $x'$ should satisfy:

$$\forall i \in \mathcal{A}(Wx + b), j \in [l] \setminus \mathcal{A}(Wx + b), W_i x' + b_i \geq (\text{or } >) W_j x' + b_j.$$

Notice that the inequality $W_i x' + b_i \geq (\text{or } >) W_j x' + b_j$ denotes a half hyperplane $(W_i - W_j)x' + (b_i - b_j) \geq (\text{or } >) 0$. Thus, the set $\{x' \in \mathbb{R}^m \mid \mathcal{A}(Wx + b) = \mathcal{A}(Wx' + b)\}$ is convex since it is an intersection of half hyperplanes. $\square$

**Lemma 2** (Different patterns of input points with small angle imply different patterns of input points with large angle). *Let $\alpha \in (0, 1)$. Given an arbitrary weight matrix $W \in \mathbb{R}^{l \times m}$, a bias vector $b = 0$, and a vector $x \in \mathbb{R}^m$ with $\|x\|_2 = 1$, if every vector $x' \in \mathbb{R}^m$ with $\|x'\|_2 = 1$ and $\langle x, x' \rangle = \alpha$ satisfies $\mathcal{A}(Wx + b) \neq \mathcal{A}(Wx' + b)$, then for any $x'' \in \mathbb{R}^m$ with $\|x''\|_2 = 1$ and $\langle x, x'' \rangle < \alpha$, it satisfies $\mathcal{A}(Wx + b) \neq \mathcal{A}(Wx'' + b)$.*

*Proof.* We draw a line between $x$ and $x''$. There must be a point $x^* \in \mathbb{R}^m$ on the line and $\langle x, x' \rangle = \alpha$, where $x' = x^*/\|x^*\|_2$. Since $b = 0$, we have $\mathcal{A}(Wx^* + b) = \mathcal{A}(Wx' + b) \neq \mathcal{A}(Wx + b)$. Since $x^*$ is on the line between $x$ and $x''$, we have $\mathcal{A}(Wx'' + b) \neq \mathcal{A}(Wx + b)$ by convexity (see Lemma 1). $\square$

**Lemma 3** (A sufficient condition for different patterns). *Consider two vectors $y \in \mathbb{R}^l$ and $y' \in \mathbb{R}^l$. If $\exists i \in \mathcal{A}(y), j \in [l] \setminus \mathcal{A}(y)$ such that $y'_i < y'_j$, then $\mathcal{A}(y) \neq \mathcal{A}(y')$.*

*Proof.* Suppose $\mathcal{A}(y) = \mathcal{A}(y')$. We have $i \in \mathcal{A}(y')$. It means that $y'_i$ is one of the top-$k$ largest values among all entries of $y'$. Thus $y'_j$ is also one of the top-$k$ largest values, and $j$ should be in $\mathcal{A}(y')$ which leads to a contradiction. $\square$

In the remaining parts, we will assume that each entry of the weight matrix $W \in \mathbb{R}^{l \times m}$ is a standard random Gaussian variable.

**Lemma 4** (Upper bound of the entires of $W$). *Consider a matrix $W \in \mathbb{R}^{l \times m}$ where each entry is a random variable with standard Gaussian distribution $N(0, 1)$. With probability at least $0.99$, $\forall i \in [l]$, $\|W_i\|_2 \leq 10\sqrt{ml}$.*

*Proof.* Consider a fixed $i \in [l]$. We have $\mathbb{E}[\|W_i\|_2^2] = m$. By Markov's inequality, we have $\Pr[\|W_i\|_2^2 > 100ml] \leq 0.01/l$. By taking union bound over all $i \in [l]$, with probability at least $0.99$, we have $\forall i \in [l], \|W_i\|_2 \leq 10\sqrt{ml}$. $\square$

**Lemma 5** (Two vectors may have different activation patterns with a good probability). *Consider a matrix $W \in \mathbb{R}^{l \times m}$ where each entry is a random variable with standard Gaussian distribution $N(0, 1)$. Let $\gamma \in (0, 0.48)$ be the sparsity ratio of the activation, i.e., $\gamma = k/l$. For any two vectors $x, x' \in \mathbb{R}^m$ with $\|x\|_2 = \|x'\|_2 = 1$ and $\langle x, x' \rangle = \alpha$ for some arbitrary $\alpha \in (0.5, 1)$, with probability at least $1 - 2^{-\Theta((1/\alpha^2 - 1)\gamma l)}$, $\mathcal{A}(Wx) \neq \mathcal{A}(Wx')$ and $\exists i \in \mathcal{A}(Wx), j \in [l] \setminus \mathcal{A}(Wx)$ such that*

$$W_i x' < W_j x' - \frac{\sqrt{1 - \alpha^2}}{24\alpha} \cdot \sqrt{2\pi}.$$

*Proof.* Consider arbitrary two vectors $x, x' \in \mathbb{R}^m$ with $\|x\|_2 = \|x'\|_2 = 1$ and $\langle x, x' \rangle = \alpha$. We can find an orthogonal matrix $Q \in \mathbb{R}^{m \times m}$ such that $\tilde{x} := Qx = (1, 0, 0, \cdots, 0)^\top \in \mathbb{R}^m$ and $\tilde{x}' := Qx' = (\alpha, \sqrt{1 - \alpha^2}, 0, 0, \cdots, 0)^\top \in \mathbb{R}^m$. Let $\tilde{W} = WQ^\top$. Then we have $\tilde{W}\tilde{x} = Wx$ and $\tilde{W}\tilde{x}' = Wx'$. Thus, we only need to analyze the activation patterns of $\tilde{W}\tilde{x}$ and $\tilde{W}\tilde{x}'$. Since $Q^\top$ is an orthogonal matrix and each entry of $W$ is an i.i.d. random variable with standard Gaussian distribution $N(0, 1)$, $\tilde{W} = WQ^\top$ is also a random matrix where each entry is an i.i.d. random variable with standard Gaussian distribution $N(0, 1)$. Let the entries in the first column of $\tilde{W}$ be $X_1, X_2, \cdots, X_l$ and let the entries in the second column of $\tilde{W}$ be $Y_1, Y_2, \cdots, Y_l$. Then we have

$$Wx = \tilde{W}\tilde{x} = \begin{pmatrix} X_1 \\ X_2 \\ \cdots \\ X_l \end{pmatrix}, \quad Wx' = \tilde{W}\tilde{x}' = \begin{pmatrix} \alpha X_1 + \sqrt{1 - \alpha^2}Y_1 \\ \alpha X_2 + \sqrt{1 - \alpha^2}Y_2 \\ \cdots \\ \alpha X_l + \sqrt{1 - \alpha^2}Y_l \end{pmatrix}. \tag{6}$$

We set $\varepsilon = \sqrt{1 - \alpha^2}/(96\alpha)$ and define $R_1' < R_1 < R_2 < R_2'$ as follows:

$$\Pr_{X \sim N(0,1)}[X \geq R_2'] = (1 - 2\varepsilon)\gamma, \tag{7}$$

$$\Pr_{X \sim N(0,1)}[X \geq R_2] = (1 - \varepsilon)\gamma, \tag{8}$$

$$\Pr_{X \sim N(0,1)}[X \geq R_1] = (1 + \varepsilon)\gamma, \tag{9}$$

$$\Pr_{X \sim N(0,1)}[X \geq R_1'] = (1 + 2\varepsilon)\gamma. \tag{10}$$

Since $\gamma < 0.48$ and $\varepsilon \leq 0.02$, we have $(1 + 2\varepsilon)\gamma < 0.5$. It implies $0 < R_1' < R_1 < R_2 < R_2'$.

**Claim 3.**
$$R_2' - R_1' \leq 8\varepsilon\sqrt{2\pi}.$$

*Proof.* By Equation (7) and Equation (10),
$$\Pr_{X \sim N(0,1)}[R_1' \leq X \leq R_2'] = 4\varepsilon\gamma.$$

Due to the density function of standard Gaussian distribution, we have

$$\frac{1}{\sqrt{2\pi}} \int_{R_1'}^{R_2'} e^{-t^2/2} \mathrm{d}t = \Pr_{X \sim N(0,1)}[R_1' \leq X \leq R_2'] = 4\varepsilon\gamma.$$

Since $R_2' \geq R_1' \geq 0$, we have $\forall t \in [R_1', R_2']$, $e^{-t^2/2} \geq e^{-R_2'^2/2}$. Thus,

$$\frac{1}{\sqrt{2\pi}} \cdot e^{-R_2'^2/2}(R_2' - R_1') = \frac{1}{\sqrt{2\pi}} \cdot e^{-R_2'^2/2} \int_{R_1'}^{R_2'} 1 \mathrm{d}t \leq \frac{1}{\sqrt{2\pi}} \int_{R_1'}^{R_2'} e^{-t^2/2} \mathrm{d}t = 4\varepsilon\gamma.$$

By the tail bound of Gaussian distribution, we have

$$\Pr_{X \sim N(0,1)}[X \geq R_2'] \leq e^{-R_2'^2/2}.$$

By combining with Equation (7), we have

$$(1 - 2\varepsilon)\gamma \cdot \frac{1}{\sqrt{2\pi}}(R_2' - R_1')$$
$$= \Pr_{X \sim N(0,1)}[X \geq R_2'] \cdot \frac{1}{\sqrt{2\pi}}(R_2' - R_1')$$
$$\leq e^{-R_2'^2/2} \cdot \frac{1}{\sqrt{2\pi}}(R_2' - R_1')$$
$$\leq 4\varepsilon\gamma,$$

which implies

$$R_2' - R_1' \leq \frac{4\varepsilon}{1 - 2\varepsilon}\sqrt{2\pi} \leq 8\varepsilon\sqrt{2\pi},$$

where the last inequality follows from $1 - 2\varepsilon \geq 0.5$. $\square$

**Claim 4.**

$$\Pr_{X_1,X_2,\cdots,X_l}\left[\sum_{i=1}^{l}\mathbf{1}(X_i \geq R_2) \geq (1 - \varepsilon/2)\gamma l\right] \leq e^{-\varepsilon^2\gamma l/24} \tag{11}$$

$$\Pr_{X_1,X_2,\cdots,X_l}\left[\sum_{i=1}^{l}\mathbf{1}(X_i \geq R_1) \leq (1 + \varepsilon/2)\gamma l\right] \leq e^{-\varepsilon^2\gamma l/18} \tag{12}$$

$$\Pr_{X_1,X_2,\cdots,X_l}\left[\sum_{i=1}^{l}\mathbf{1}(R_2' \geq X_i \geq R_2) \leq \varepsilon\gamma l/2\right] \leq e^{-\varepsilon\gamma l/8} \tag{13}$$

$$\Pr_{X_1,X_2,\cdots,X_l}\left[\sum_{i=1}^{l}\mathbf{1}(R_1 \geq X_i \geq R_1') \leq \varepsilon\gamma l/2\right] \leq e^{-\varepsilon\gamma l/8} \tag{14}$$

*Proof.* For $i \in [l]$, we have $\mathbb{E}[\mathbf{1}(X_i \geq R_2)] = \Pr[X_i \geq R_2] = (1 - \varepsilon)\gamma$ by Equation (8). By Chernoff bound, we have

$$\Pr\left[\sum_{i=1}^{l}\mathbf{1}(X_i \geq R_2) \geq (1 + \varepsilon/2) \cdot (1 - \varepsilon)\gamma l\right] \leq e^{-(\varepsilon/2)^2(1-2\varepsilon)\gamma l/3}.$$

Since $\varepsilon \leq 0.02$,

$$\Pr\left[\sum_{i=1}^{l}\mathbf{1}(X_i \geq R_2) \geq (1 - \varepsilon/2)\gamma l\right] \leq e^{-\varepsilon^2\gamma l/24}.$$

We have $\mathbb{E}[\mathbf{1}(X_i \geq R_1)] = \Pr[X_i \geq R_1] = (1 + \varepsilon)\gamma$ by Equation (9). By Chernoff bound, we have

$$\Pr\left[\sum_{i=1}^{l}\mathbf{1}(X_i \geq R_1) \leq (1 - \varepsilon/3) \cdot (1 + \varepsilon)\gamma l\right] \leq e^{-(\varepsilon/3)^2(1+\varepsilon)\gamma l/2}.$$

Thus,

$$\Pr\left[\sum_{i=1}^{l}\mathbf{1}(X_i \geq R_i) \leq (1 + \varepsilon/2)\gamma l\right] \leq e^{-\varepsilon^2\gamma l/18}$$

We have $\mathbb{E}\left[\mathbf{1}(R_2' \geq X_i \geq R_2)\right] = \Pr[R_2' \geq X_i \geq R_2] = \varepsilon\gamma$ by Equation (7) and Equation (8). By Chernoff bound, we have

$$\Pr\left[\sum_{i=1}^{l}\mathbf{1}(R_2' \geq X_i \geq R_2) \leq 1/2 \cdot \varepsilon\gamma l\right] \leq e^{-\varepsilon\gamma l/8}$$

Similarly, we have $\mathbb{E}[\mathbf{1}(R_1 \geq X_i \geq R_1')] = \Pr[R_1 \geq X_i \geq R_1'] = \varepsilon\gamma$ by Equation (9) and Equation (10). By chernoff bound, we have

$$\Pr_{X_1,X_2,\cdots,X_l}\left[\sum_{i=1}^{l}\mathbf{1}(R_1 \geq X_i \geq R_1') \leq 1/2 \cdot \varepsilon\gamma l\right] \leq e^{-\varepsilon\gamma l/8}$$

□

Equation (11) says that, with high probability, $\forall i \in [l]$ with $X_i \geq R_2$, it has $i \in \mathcal{A}(Wx)$. Equation (12) says that, with high probability, $\forall i \in [l]$ with $X_i \leq R_1$, it has $i \notin \mathcal{A}(Wx)$. Equation (14) (Equation (13)) says that, with high probability, there are many $i \in [l]$ such that $W_i x \in [R_1', R_1]$ ($W_i x \in [R_2, R_2']$).

Let $\mathcal{E} = \mathcal{E}_1 \wedge \mathcal{E}_2 \wedge \mathcal{E}_3 \wedge \mathcal{E}_4$, where

- $\mathcal{E}_1$: $\sum_{i=1}^{l}\mathbf{1}(X_i \geq R_2) \leq (1 - \varepsilon/2)\gamma l$,
- $\mathcal{E}_2$: $\sum_{i=1}^{l}\mathbf{1}(X_i \geq R_1) \geq (1 + \varepsilon/2)\gamma l$,

- $\mathcal{E}_3$: $\sum_{i=1}^{l} \mathbf{1}(R_1 \geq X_i \geq R_1') \geq \varepsilon\gamma l/2$,

- $\mathcal{E}_4$: $\sum_{i=1}^{l} \mathbf{1}(R_2' \geq X_i \geq R_2) \geq \varepsilon\gamma l/2$.

According to Equation (11), Equation (12), Equation (13) and Equation (14), the probability that $\mathcal{E}$ happens is at least

$$1 - 4e^{-\varepsilon^2\gamma l/24} \tag{15}$$

by union bound over $\bar{\mathcal{E}}_1, \bar{\mathcal{E}}_2, \bar{\mathcal{E}}_3, \bar{\mathcal{E}}_4$.

**Claim 5.** *Condition on $\mathcal{E}$, the probability that $\exists i \in [l]$ with $X_i \in [R_2, R_2']$ such that $Y_i < -\alpha/\sqrt{1-\alpha^2} \cdot 16\varepsilon\sqrt{2\pi}$ is at least*

$$1 - \left(16\varepsilon \cdot \frac{\alpha}{\sqrt{1-\alpha^2}} + \frac{1}{2}\right)^{\varepsilon\gamma l/2}.$$

*Proof.* For a fixed $i \in [l]$,

$$
\begin{aligned}
\Pr\left[Y_i \geq -\alpha/\sqrt{1-\alpha^2} \cdot 16\varepsilon\sqrt{2\pi}\right] &= \int_{-\alpha/\sqrt{1-\alpha^2}\cdot 16\varepsilon\sqrt{2\pi}}^{0} \frac{1}{\sqrt{2\pi}} e^{-t^2/2}\mathrm{d}t + \frac{1}{2} \\
&\leq \frac{1}{\sqrt{2\pi}} \cdot \alpha/\sqrt{1-\alpha^2} \cdot 16\varepsilon\sqrt{2\pi} + \frac{1}{2} \\
&= 16\varepsilon \cdot \frac{\alpha}{\sqrt{1-\alpha^2}} + \frac{1}{2}.
\end{aligned}
$$

Thus, according to event $\mathcal{E}_4$, we have

$$\Pr\left[\forall i \text{ with } X_i \in [R_2, R_2'], Y_i \geq -\alpha/\sqrt{1-\alpha^2} \cdot 16\varepsilon\sqrt{2\pi} \mid \mathcal{E}\right] \leq \left(16\varepsilon \cdot \frac{\alpha}{\sqrt{1-\alpha^2}} + \frac{1}{2}\right)^{\varepsilon\gamma l/2}.$$

$\square$

**Claim 6.** *Condition on $\mathcal{E}$, the probability that $\exists i \in [l]$ with $X_i \in [R_1', R_1]$ such that $Y_i \geq 0$ is at least $1 - (1/2)^{\varepsilon\gamma l/2}$.*

*Proof.* For a fixed $i \in [l]$, $\Pr[Y_i \leq 0] = 1/2$. Thus, according to event $\mathcal{E}_3$, we have

$$\Pr\left[\forall i \text{ with } X_i \in [R_1', R_1], Y_i \leq 0 \mid \mathcal{E}\right] \leq (1/2)^{\varepsilon\gamma l/2}.$$

$\square$

Condition on that $\mathcal{E}$ happens. Because of $\mathcal{E}_1$, if $X_i \geq R_2$, $X_i$ must be one of the top-$k$ largest values. Due to Equation (6), we have $X_i = W_i x$. Thus, if $X_i \geq R_2$, $i \in \mathcal{A}(Wx)$. By Claim 5, with probability at least

$$1 - \left(16\varepsilon \cdot \frac{\alpha}{\sqrt{1-\alpha^2}} + \frac{1}{2}\right)^{\varepsilon\gamma l/2}, \tag{16}$$

there is $i \in \mathcal{A}(Wx)$ such that

$$
\begin{aligned}
W_i x' &= \alpha X_i + \sqrt{1-\alpha^2} Y_i \\
&\leq \alpha X_i + \sqrt{1-\alpha^2} \cdot \left(-\frac{\alpha}{\sqrt{1-\alpha^2}} \cdot 16\varepsilon\sqrt{2\pi}\right) \\
&= \alpha(X_i - 16\varepsilon\sqrt{2\pi}) \\
&\leq \alpha(R_2' - 16\varepsilon\sqrt{2\pi}), \tag{17}
\end{aligned}
$$

where the first step follows from Equation (6), the second step follows from $Y_i \leq -\alpha/\sqrt{1-\alpha^2} \cdot 16\varepsilon\sqrt{2\pi}$, and the last step follows from $X_i \in [R_2, R_2']$.

Because of $\mathcal{E}_2$ if $X_j \leq R_1$, $X_j$ should not be one of the top-$k$ largest values. Due to Equation (6), we have $X_j = W_j x$. Thus, if $X_j \leq R_1$, $j \notin \mathcal{A}(Wx)$. By Claim C.1, with probability at least

$$1 - (1/2)^{\varepsilon\gamma l/2}, \tag{18}$$

there is $j \notin \mathcal{A}(Wx)$ such that

$$W_j x' = \alpha X_j + \sqrt{1-\alpha^2} Y_j \geq \alpha X_j \geq \alpha R_1', \tag{19}$$

where the first step follows from Equation (6), the second step follows from $Y_j \geq 0$, and the last step follows from $X_j \in [R_1', R_1]$.

By Equation (19) and Equation (17), $\exists i \in \mathcal{A}(Wx), j \in [l] \setminus \mathcal{A}(Wx)$,

$$W_i x' \leq \alpha(R_2' - 16\varepsilon\sqrt{2\pi}) \leq \alpha(R_1' - 8\varepsilon\sqrt{2\pi}) \leq W_j x' - 8\alpha\varepsilon\sqrt{2\pi}$$

$$\leq W_j x' - 4\varepsilon\sqrt{2\pi} = W_j x' - \frac{\sqrt{1-\alpha^2}}{24\alpha} \cdot \sqrt{2\pi},$$

where the second step follows from Claim 3, the forth step follows from $\alpha \geq 0.5$, and the last step follows from $\varepsilon = \sqrt{1-\alpha^2}/(96\alpha)$. By Lemma 3, we can conclude $\mathcal{A}(Wx) \neq \mathcal{A}(Wx')$. By Equation (15), Equation (16), Equation (18), and union bound, the overall probability is at least

$$1 - \left( 4e^{-\varepsilon^2\gamma l/24} + \left( 16\varepsilon \cdot \frac{\alpha}{\sqrt{1-\alpha^2}} + \frac{1}{2} \right)^{\varepsilon\gamma l/2} + \left( \frac{1}{2} \right)^{\varepsilon\gamma l/2} \right)$$

$$\geq 1 - \left( 4e^{-\varepsilon^2\gamma l/24} + \left( \frac{2}{3} \right)^{\varepsilon\gamma l/2} + \left( \frac{1}{2} \right)^{\varepsilon\gamma l/2} \right)$$

$$\geq 1 - 6 \cdot \left( \frac{2}{3} \right)^{\varepsilon^2\gamma l/24}$$

$$\geq 1 - 2^{-\Theta\left( \left( \frac{1}{\alpha^2} - 1 \right)\gamma l \right)},$$

where the first and the last step follows from $\varepsilon = \sqrt{1-\alpha^2}/(96\alpha)$ □

Next, we will use a tool called $\varepsilon$-net.

**Definition 7** ($\varepsilon$-Net). *For a given set $\mathcal{S}$, if there is a set $\mathcal{N} \subseteq \mathcal{S}$ such that $\forall x \in \mathcal{S}$ there exists a vector $y \in \mathcal{N}$ such that $\|x - y\|_2 \leq \varepsilon$, then $\mathcal{N}$ is an $\varepsilon$-net of $\mathcal{S}$.*

There is a standard upper bound of the size of an $\varepsilon$-net of a unit norm ball.

**Lemma 6** (Wojtaszczyk (1996) II.E, 10). *Given a matrix $U \in \mathbb{R}^{m \times d}$, let $\mathcal{S} = \{Uy \mid \|Uy\|_2 = 1\}$. For $\varepsilon \in (0, 1)$, there is an $\varepsilon$-net $\mathcal{N}$ of $\mathcal{S}$ with $|\mathcal{N}| \leq (1 + 1/\varepsilon)^d$.*

Now we can extend above lemma to the following.

**Lemma 7** ($\varepsilon$-Net for the set of points with a certain angle). *Given a vector $x \in \mathbb{R}^m$ with $\|x\|_2 = 1$ and a parameter $\alpha \in (-1, 1)$, let $\mathcal{S} = \{x' \in \mathbb{R}^m \mid \|x'\|_2 = 1, \langle x, x' \rangle = \alpha\}$. For $\varepsilon \in (0, 1)$, there is an $\varepsilon$-net $\mathcal{N}$ of $\mathcal{S}$ with $|\mathcal{N}| \leq (1 + 1/\varepsilon)^{m-1}$.*

*Proof.* Let $U \in \mathbb{R}^{m \times (m-1)}$ have orthonormal columns and $Ux = 0$. Then $\mathcal{S}$ can be represented as

$$\mathcal{S} = \{\alpha \cdot x + \sqrt{1-\alpha^2} \cdot Uy \mid y \in \mathbb{R}^{m-1}, \|Uy\|_2 = 1\}.$$

Let

$$\mathcal{S}' = \{Uy \mid y \in \mathbb{R}^{m-1}, \|Uy\|_2 = 1\}.$$

According to Lemma 6, there is an $\varepsilon$-net $\mathcal{N}'$ of $\mathcal{S}'$ with size $|\mathcal{N}'| \leq (1 + 1/\varepsilon)^{m-1}$. We construct $\mathcal{N}$ as following:

$$\mathcal{N} = \{\alpha \cdot x + \sqrt{1-\alpha^2} \cdot z \mid z \in \mathcal{N}'\}.$$

It is obvious that $|\mathcal{N}| = |\mathcal{N}'| \leq (1 + 1/\varepsilon)^{m-1}$. Next, we will show that $\mathcal{N}$ is indeed an $\varepsilon$-net of $\mathcal{S}$. Let $x'$ be an arbitrary vector from $\mathcal{S}$. Let $x' = \alpha \cdot x + \sqrt{1-\alpha^2} \cdot z$ for some $z \in \mathcal{S}'$. There is a vector $(\alpha \cdot x + \sqrt{1-\alpha^2} \cdot z') \in \mathcal{N}$ such that $z' \in \mathcal{N}'$ and $\|z - z'\|_2 \leq \varepsilon$. Thus, we have

$$\|x' - (\alpha \cdot x + \sqrt{1-\alpha^2} \cdot z')\|_2 = \sqrt{1-\alpha^2}\|z - z'\|_2 \leq \varepsilon.$$

□

**Theorem 8** (Rotating a vector a little bit may change the activation pattern). *Consider a weight matrix* $W \in \mathbb{R}^{l \times m}$ *where each entry is an i.i.d. sample drawn from the Gaussian distribution* $N(0, 1/l)$*. Let* $\gamma \in (0, 0.48)$ *be the sparsity ratio of the activation function, i.e.,* $\gamma = k/l$*. With probability at least* $0.99$*, it has* $\forall i \in [l], \|W_i\|_2 \leq 10\sqrt{m}$*. Condition on that* $\forall i \in [l], \|W_i\|_2 \leq 10\sqrt{m}$ *happens, then, for any* $x \in \mathbb{R}^m$ *and* $\alpha \in (0.5, 1)$*, if*

$$l \geq C \cdot \left( \frac{m + \log(1/\delta)}{\gamma} \cdot \frac{1}{1 - \alpha^2} \right) \cdot \log \left( \frac{m + \log(1/\delta)}{\gamma} \cdot \frac{1}{1 - \alpha^2} \right)$$

*for a sufficiently large constant* $C$*, with probability at least* $1 - \delta \cdot 2^{-m}$*,* $\forall x' \in \mathbb{R}^m$ *with* $\frac{\langle x, x' \rangle}{\|x\|_2 \|x'\|_2} \leq \alpha$*,* $\mathcal{A}(Wx) \neq \mathcal{A}(Wx')$*.*

*Proof.* Notice that the scale of $W$ does not affect the activation pattern of $Wx$ for any $x \in \mathbb{R}^m$. Thus, we assume that each entry of $W$ is a standard Gaussian random variable in the remaining proof, and we will instead condition on $\forall i \in [l], \|W_i\|_2 \leq 10\sqrt{ml}$. The scale of $x$ or $x'$ will not affect $\frac{\langle x, x' \rangle}{\|x\|_2 \|x'\|_2}$. It will not affect the activation pattern either. Thus, we assume $\|x\|_2 = \|x'\|_2 = 1$.

By Lemma 4, with probability at least $0.99$, we have $\forall i \in [l], \|W_i\|_2 \leq 10\sqrt{ml}$.

Let

$$\mathcal{S} = \{ y \in \mathbb{R}^m \mid \|y\|_2 = 1, \langle x, y \rangle = \alpha \}.$$

Set

$$\varepsilon = \frac{\sqrt{2\pi(1 - \alpha^2)}}{720\alpha\sqrt{ml}}.$$

By Lemma 7, there is an $\varepsilon$-net $\mathcal{N}$ of $\mathcal{S}$ such that

$$|\mathcal{N}| \leq \left( 1 + \frac{720\alpha\sqrt{ml}}{\sqrt{2\pi(1 - \alpha^2)}} \right)^m.$$

By Lemma 5, for any $y \in \mathcal{N}$, with probability at least

$$1 - 2^{-\Theta((1/\alpha^2 - 1)\gamma l)},$$

$\exists i \in \mathcal{A}(Wx), j \in [l] \setminus \mathcal{A}(Wx)$ such that

$$W_i y < W_j y - \frac{\sqrt{1 - \alpha^2}}{24\alpha} \cdot \sqrt{2\pi}.$$

By taking union bound over all $y \in \mathcal{N}$, with probability at least

$$1 - |\mathcal{N}| \cdot 2^{-\Theta((1/\alpha^2 - 1)\gamma l)}$$

$$\geq 1 - \left( 1 + \frac{720\alpha\sqrt{ml}}{\sqrt{2\pi(1 - \alpha^2)}} \right)^m 2^{-\Theta((\frac{1}{\alpha^2} - 1)\gamma l)}$$

$$\geq 1 - \left( 1000 \cdot \frac{\sqrt{ml}}{\sqrt{1 - \alpha^2}} \right)^m 2^{-\Theta((\frac{1}{\alpha^2} - 1)\gamma l)}$$

$$\geq 1 - \left( 1000 \cdot \frac{\sqrt{ml}}{\sqrt{1 - \alpha^2}} \right)^m 2^{-C' \cdot (\frac{1}{\alpha^2} - 1)\gamma \cdot \frac{m + \log(1/\delta)}{\gamma} \cdot \frac{\alpha^2}{1 - \alpha^2} \cdot \log\left( \frac{ml}{1 - \alpha^2} \right)} \quad // \ C' \text{ is a sufficiently large constant}$$

$$= 1 - \left( 1000 \cdot \frac{\sqrt{ml}}{\sqrt{1 - \alpha^2}} \right)^m 2^{-C' \cdot (m + \log(1/\delta)) \cdot \log\left( \frac{ml}{1 - \alpha^2} \right)}$$

$$\geq 1 - \delta \cdot 2^{-m},$$

the following event $\mathcal{E}'$ happens: $\forall y \in \mathcal{N}, \exists i \in \mathcal{A}(Wx), j \in [l] \setminus \mathcal{A}(Wx)$ such that

$$W_i y < W_j y - \frac{\sqrt{1 - \alpha^2}}{24\alpha} \cdot \sqrt{2\pi}.$$

In the remaining of the proof, we will condition on the event $\mathcal{E}'$. Consider $y' \in \mathcal{S}$. Since $\mathcal{N}$ is an $\varepsilon$-net of $\mathcal{S}$, we can always find a $y \in \mathcal{N}$ such that

$$\|y - y'\|_2 \leq \varepsilon = \frac{\sqrt{2\pi(1 - \alpha^2)}}{720\alpha\sqrt{ml}}.$$

Since event $\mathcal{E}'$ happens, we can find $i \in \mathcal{A}(Wx)$ and $j \in [l] \setminus \mathcal{A}(Wx)$ such that

$$W_i y < W_j y - \frac{\sqrt{1 - \alpha^2}}{24\alpha} \cdot \sqrt{2\pi}.$$

Then, we have

$$\begin{aligned}
W_i y' &= W_i y + W_i(y' - y) \\
&< W_j y - \frac{\sqrt{1 - \alpha^2}}{24\alpha} \cdot \sqrt{2\pi} + \|W_i\|_2 \|y' - y\|_2 \\
&\leq W_j y - \frac{\sqrt{1 - \alpha^2}}{24\alpha} \cdot \sqrt{2\pi} + 10\sqrt{ml} \cdot \frac{\sqrt{2\pi(1 - \alpha^2)}}{720\alpha\sqrt{ml}} \\
&= W_j y - \frac{\sqrt{1 - \alpha^2}}{36\alpha} \cdot \sqrt{2\pi} \\
&= W_j y' + W_j(y - y') - \frac{\sqrt{1 - \alpha^2}}{36\alpha} \cdot \sqrt{2\pi} \\
&\leq W_j y' + \|W_j\|_2 \|y - y'\|_2 - \frac{\sqrt{1 - \alpha^2}}{36\alpha} \cdot \sqrt{2\pi} \\
&\leq W_j y' + 10\sqrt{ml} \cdot \frac{\sqrt{2\pi(1 - \alpha^2)}}{720\alpha\sqrt{ml}} - \frac{\sqrt{1 - \alpha^2}}{36\alpha} \cdot \sqrt{2\pi} \\
&\leq W_j y' - \frac{\sqrt{1 - \alpha^2}}{72\alpha} \cdot \sqrt{2\pi},
\end{aligned}$$

where the second step follows from $W_i y < W_j y - \sqrt{1 - \alpha^2}/(24\alpha) \cdot \sqrt{2\pi}$ and $W_i(y' - y) \leq \|W_i\|_2 \|y' - y\|_2$, the third step follows from $\|W_i\|_2 \leq 10\sqrt{ml}$ and $\|y' - y\|_2 \leq \sqrt{2\pi(1 - \alpha^2)}/(720\alpha\sqrt{ml})$, the sixth step follows from $W_j(y - y') \leq \|W_j\|_2 \|y - y'\|_2$, and the seventh step follows from $\|W_i\|_2 \leq 10\sqrt{ml}$ and $\|y' - y\|_2 \leq \sqrt{2\pi(1 - \alpha^2)}/(720\alpha\sqrt{ml})$.

By Lemma 3, we know that $\mathcal{A}(Wx) \neq \mathcal{A}(Wy')$. Thus, $\forall y' \in \mathbb{R}^m$ with $\|y'\|_2 = 1$ and $\langle x, y' \rangle = \alpha$, we have $\mathcal{A}(Wx) \neq \mathcal{A}(Wy')$ conditioned on $\mathcal{E}'$. By Lemma 2, we can conclude that $\forall x' \in \mathbb{R}^m$ with $\|x'\|_2 = 1$ and $\langle x, x' \rangle \leq \alpha$, we have $\mathcal{A}(Wx) \neq \mathcal{A}(Wx')$ conditioned on $\mathcal{E}'$. $\qquad\square$

**Theorem 9** (A formal version of Theorem 1). *Consider a weight matrix $W \in \mathbb{R}^{l \times m}$ where each entry is an i.i.d. sample drawn from the Gaussian distribution $N(0, 1/l)$. Let $\gamma \in (0, 0.48)$ be the sparsity ratio of the activation function, i.e., $\gamma = k/l$. With probability at least $0.99$, it has $\forall i \in [l], \|W_i\|_2 \leq 10\sqrt{m}$. Condition on that $\forall i \in [l], \|W_i\|_2 \leq 10\sqrt{m}$ happens, then, for any $x \in \mathbb{R}^m$, if*

$$l \geq C \cdot \left( \frac{m + \log(1/\delta)}{\gamma} \cdot \frac{1}{\beta} \right) \cdot \log\left( \frac{m + \log(1/\delta)}{\gamma} \cdot \frac{1}{\beta} \right)$$

*for some $\beta \in (0, 1)$ and a sufficiently large constant $C$, with probability at least $1 - \delta \cdot 2^{-m}$, $\forall x' \in \mathbb{R}^m$ with $\|\Delta x\|_2^2 / \|x\|_2^2 \geq \beta$, $\mathcal{A}(Wx) \neq \mathcal{A}(Wx')$, where $x' = c \cdot (x + \Delta x)$ for some scaler $c$, and $\Delta x$ is perpendicular to $x$.*

*Proof.* If $\langle x, x' \rangle \leq 0$, then the statement follows from Theorem 8 directly. In the following, we consider the case $\langle x, x' \rangle > 0$. If $\|\Delta x\|_2 / \|x\|_2^2 \geq \beta$,

$$\begin{aligned}
&\frac{\langle x, x' \rangle^2}{\|x\|_2^2 \|x'\|_2^2} \\
&= \frac{c^2 \|x\|_2^4}{\|x\|_2^2 (c^2 (\|x\|_2^2 + \|\Delta x\|_2^2))} = \frac{\|x\|_2^2}{\|x\|_2^2 + \|\Delta x\|_2^2} \\
&\leq \frac{\|x\|_2^2}{\|x\|_2^2 + \beta \|x\|_2^2} \leq \frac{1}{1 + \beta}.
\end{aligned}$$

Thus, we have the bounds:

$$\frac{1}{1 - \frac{\langle x, x' \rangle^2}{\|x\|_2^2 \|x'\|_2^2}} \le \frac{1}{\beta} + 1 \le O\left(\frac{1}{\beta}\right).$$

By Theorem 8, we conclude the proof. $\square$

**Example 1.** *Suppose that the training data contains $N$ points $x_1, x_2, \cdots, x_N \in \mathbb{R}^m$ ($m \ge \Omega(\log N)$), where each entry of $x_i$ for $i \in [N]$ is an i.i.d. Bernoulli random variable, i.e., each entry is 1 with some probability $p \in (100 \log(N)/m, 0.5)$ and 0 otherwise. Consider a weight matrix $W \in \mathbb{R}^{l \times m}$ where each entry is an i.i.d. sample drawn from the Gaussian distribution $N(0, 1/l)$. Let $\gamma \in (0, 0.48)$ be the sparsity ratio of the activation function, i.e., $\gamma = k/l$. If $l \ge \Omega(m/\gamma \cdot \log(m/\gamma))$, then with probability at least 0.9, $\forall i, j \in [N]$, the activation pattern of $W x_i$ and $W x_j$ are different, i.e., $\mathcal{A}(W x_i) \ne \mathcal{A}(W x_j)$.*

*Proof.* Firstly, let us bound $\|x_i\|_2$. We have $\mathbb{E}[\|x_i\|_2^2] = \mathbb{E}[\sum_{t=1}^m x_{i,t}] = pm$. By Bernstein inequality, we have

$$\Pr\left[\left|\sum_{t=1}^m x_{i,t} - pm\right| > \frac{1}{10}pm\right] \le 2e^{-\frac{(pm/10)^2/2}{pm + \frac{1}{3} \cdot \frac{1}{10}pm}} \le 0.01/N.$$

Thus, by taking union bound over all $i \in [N]$, with probability at least 0.99, $\forall i \in [N]$, $\sqrt{0.9pm} \le \|x_i\|_2 \le \sqrt{1.1pm}$.

Next we consider $\langle x_i, x_j \rangle$. Notice that $\mathbb{E}[\langle x_i, x_j \rangle] = \mathbb{E}[\sum_{t=1}^m x_{i,t} x_{j,t}] = p^2 m$. There are two cases.

**Case 1** ($p^2 m > 20 \log N$). By Bernstein inequality, we have

$$\Pr\left[|\langle x_i, x_j \rangle - p^2 m| > \frac{1}{2}p^2 m\right] \le 2e^{-\frac{(p^2 m/2)^2/2}{p^2 m + \frac{1}{3}\frac{1}{2}p^2 m}} = 2e^{-\frac{3}{28}p^2 m} \le 0.01/N^2.$$

By taking union bound over all pairs of $i, j$, with probability at least 0.99, $\forall i \ne j$, $\langle x_i, x_j \rangle \le \frac{3}{2}p^2 m$. Since $\|x_i\|_2, \|x_j\|_2 \ge \sqrt{0.9pm}$, we have

$$\frac{\langle x_i, x_j \rangle}{\|x_i\|_2 \|x_j\|_2} \le \frac{3p^2 m/2}{0.9pm} = \frac{5}{3}p \le \frac{5}{6}.$$

**Case 2** ($p^2 m \le 20 \log N$). By Bernstein inequality, we have

$$\Pr\left[|\langle x_i, x_j \rangle - p^2 m| > 10 \log N\right] \le 2e^{-\frac{(10 \log N)^2/2}{p^2 m + \frac{1}{3} \cdot 10 \log N}} \le 0.01/N^2.$$

By taking union bound over all pairs of $i, j$, with probability at least 0.99, $\forall i \ne j$, $\langle x_i, x_j \rangle \le 10 \log N$, Since $\|x_i\|_2, \|x_j\|_2 \ge \sqrt{0.9pm} \ge \sqrt{90 \log N}$, we have

$$\frac{\langle x_i, x_j \rangle}{\|x_i\|_2 \|x_j\|_2} \le \frac{10 \log N}{90 \log N} = \frac{1}{9}.$$

Thus, with probability at least 0.98, we have $\forall i \ne j$, $\langle x_i, x_j \rangle/(\|x_i\|_2 \|x_j\|_2) \le 5/6$. By Theorem 8, with probability at least 0.99, $\forall q \in [l]$, $\|W_q\|_2 \le 10\sqrt{m}$. Condition on this event, and since $\forall i \ne j$ we have $\langle x_i, x_j \rangle/(\|x_i\|_2 \|x_j\|_2) \le 5/6$, by Theorem 8 again and union bound over all $i \in [N]$, with probability at least 0.99, $\forall i \ne j$, $\mathcal{A}(W x_i) \ne \mathcal{A}(W x_j)$. $\square$

## C.2 DISJOINTNESS OF ACTIVATION PATTERNS OF DIFFERENT INPUT POINTS

Let $X_1, X_2, \cdots, X_m$ be i.i.d. random variables drawn from the standard Gaussian distribution $N(0, 1)$. Let $Z = \sum_{i=1}^m X_i^2$. We use the notation $\chi_m^2$ to denote the distribution of $Z$. If $m$ is clear in the context, we just use $\chi^2$ for short.

**Lemma 8** (A property of $\chi^2$ distribution). *Let $Z$ be a random variable with $\chi_m^2$ $m$ ($m \ge 2$) distribution. Given arbitrary $\varepsilon, \eta \in (0, 1)$, if $R$ is sufficiently large then*

$$\Pr[Z \ge (1 + \varepsilon)R]/\Pr[(1 + \varepsilon)R \ge Z \ge R] \le \eta.$$

*Proof.* Let $R$ be a sufficiently large number such that:

- $e^{\varepsilon R/2} \geq \frac{4}{\varepsilon}$.
- $e^{\varepsilon R/8} \geq R^{m/2-1}$.
- $e^{\varepsilon R/4} \geq \frac{16}{9} \cdot \frac{1}{\eta}$.

Let $\xi = \varepsilon/4$. By the density function of $\chi^2$ distribution, we have

$$\Pr[R \leq Z \leq (1+\varepsilon)R] = \frac{1}{2^{m/2}\Gamma(m/2)} \int_R^{(1+\varepsilon)R} t^{m/2-1} e^{-t/2} \mathrm{d}t,$$

and

$$\Pr[Z \geq (1+\varepsilon)R] = \frac{1}{2^{m/2}\Gamma(m/2)} \int_{(1+\varepsilon)R}^{\infty} t^{m/2-1} e^{-t/2} \mathrm{d}t,$$

where $\Gamma(\cdot)$ is the Gamma function, and for integer $m/2$, $\Gamma(m/2) = (m/2-1)(m/2-2)\cdots 2 \cdot 1 = (m/2-1)!$. By our choice of $R$, we have

$$\begin{aligned}
\Pr[R \leq Z \leq (1+\varepsilon)R] &\geq \frac{1}{2^{m/2}\Gamma(m/2)} \int_R^{(1+\varepsilon)R} e^{-t/2} \mathrm{d}t \\
&= \frac{1}{2^{m/2}\Gamma(m/2)} \cdot 2 \left( e^{-R/2} - e^{-(1+\varepsilon)R/2} \right) \\
&\geq \frac{1}{2^{m/2}\Gamma(m/2)} \cdot 2(1-\xi) \cdot e^{-R/2},
\end{aligned}$$

where the first step follows from $\forall t \geq R$, $t^{m/2-1} \geq 1$, and the third step follows from

$$\frac{e^{-(1+\varepsilon)R/2}}{e^{-R/2}} = e^{-\varepsilon R/2} \leq \xi.$$

We also have:

$$\begin{aligned}
\Pr[Z \geq (1+\varepsilon)R] &\leq \frac{1}{2^{m/2}\Gamma(m/2)} \int_{(1+\varepsilon)R}^{+\infty} e^{-(1-\xi)t/2} \mathrm{d}t \\
&= \frac{1}{2^{m/2}\Gamma(m/2)} \cdot \frac{2}{1-\xi} \cdot e^{-(1-\xi)(1+\varepsilon)R/2} \\
&\leq \frac{1}{2^{m/2}\Gamma(m/2)} \cdot \frac{2}{1-\xi} \cdot e^{-(1+\varepsilon/2)R/2},
\end{aligned}$$

where the first step follos from $\forall t \geq R$, $t^{m/2-1} \leq e^{\xi t/2}$, and the third step follows from $(1-\xi)(1+\varepsilon) \geq (1+\varepsilon/2)$.

Thus, we have

$$\frac{\Pr[Z \geq (1+\varepsilon)R]}{\Pr[(1+\varepsilon)R \geq Z \geq R]} \leq \frac{1}{(1-\xi)^2} e^{-\varepsilon R/4} \leq \frac{16}{9} e^{-\varepsilon R/4} \leq \eta.$$

$\square$

**Lemma 9.** *Consider $x, y, z \in \mathbb{R}^m$. If $\frac{\langle x, y \rangle}{\|x\|_2 \|y\|_2} \leq \alpha$, $\frac{\langle x, z \rangle}{\|x\|_2 \|z\|_2} \geq \beta$ for some $\alpha, \beta \geq 0$, then $\frac{\langle y, z \rangle}{\|y\|_2 \|z\|_2} \leq \alpha + \sqrt{1-\beta^2}$. Furthermore, if $\beta = \frac{2+\alpha+\sqrt{2-\alpha^2}}{4}$, then $\frac{\langle y, z \rangle}{\|y\|_2 \|z\|_2} \leq (1-\varepsilon_\alpha)\beta$, where $\varepsilon_\alpha \in (0, 1)$ only depends on $\alpha$.*

*Proof.* Without loss of generality, we suppose $\|x\|_2 = \|y\|_2 = \|z\|_2 = 1$. We can decompose $y$ as $ax + y'$ where $y'$ is perpendicular to $x$. We can decompose $z$ as $b_1 x + b_2 y'/\|y'\|_2 + z'$ where $z'$ is perpendicular to both $x$ and $y'$. Then we have:

$$\langle y, z \rangle = ab_1 + b_2 \|y'\|_2 \leq \alpha + \sqrt{1-\beta^2},$$

where the last inequality follows from $0 \leq b_1 \leq 1, a \leq \alpha$, and $b_2 \leq \sqrt{1 - b_1^2} \leq \sqrt{1 - \beta^2}$, $0 \leq \|y'\|_2 \leq 1$.

By solving $\beta \geq \alpha + \sqrt{1 - \beta^2}$, we can get $\beta \geq \frac{\alpha + \sqrt{2 - \alpha^2}}{2}$. Thus, if we set

$$\beta = \frac{1 + \frac{\alpha + \sqrt{2 - \alpha^2}}{2}}{2},$$

$\beta$ should be strictly larger than $\alpha + \sqrt{1 - \beta^2}$, and the gap only depends on $\alpha$. □

**Lemma 10.** *Give $x \in \mathbb{R}^m$, let $y \in \mathbb{R}^m$ be a random vector, where each entry of $y$ is an i.i.d. sample drawn from the standard Gaussian distribution $N(0, 1)$. Given $\beta \in (0.5, 1)$, $\Pr[\langle x, y \rangle / (\|x\|_2 \|y\|_2) \geq \beta] \geq 1/(1 + 1/\sqrt{2(1 - \beta)})^m$.*

*Proof.* Without loss of generality, we can assume $\|x\|_2 = 1$. Let $y' = y/\|y\|_2$. Since each entry of $y$ is an i.i.d. Gaussian variable, $y'$ is a random vector drawn uniformly from a unit sphere. Notice that if $\langle x, y' \rangle \geq \beta$, then $\|x - y'\|_2 \leq \sqrt{2(1 - \beta)}$. Let $C = \{z \in \mathbb{R}^m \mid \|z\|_2 = 1, \|z - x\|_2 \leq \sqrt{2(1 - \beta)}\}$ be a cap, and let $\mathcal{S} = \{z \in \mathbb{R}^m \mid \|z\|_2 = 1\}$ be the unit sphere. Then we have

$$\Pr[\langle x, y' \rangle \geq \beta] = \text{area}(C)/\text{area}(\mathcal{S}).$$

According to Lemma 6, there is an $\sqrt{2(1 - \beta)}$-net $\mathcal{N}$ with $|\mathcal{N}| \leq (1 + 1/\sqrt{2(1 - \beta)})^m$. If we put a cap centered at each point in $\mathcal{N}$, then the whole unit sphere will be covered. Thus, we can conclude

$$\Pr[\langle x, y' \rangle \geq \beta] \geq 1/(1 + 1/\sqrt{2(1 - \beta)})^m.$$

□

**Theorem 10** (A formal version of Theorem 2). *Consider $N$ data points $x_1, x_2, \cdots, x_N \in \mathbb{R}^m$ and a weight matrix $W \in \mathbb{R}^{l \times m}$ where each entry of $W$ is an i.i.d. sample drawn from the Gaussian distribution $N(0, 1/l)$. Suppose $\forall i \neq j \in [N]$, $\langle x_i, x_j \rangle / (\|x_i\|_2 \|x_j\|_2) \leq \alpha$ for some $\alpha \in (0.5, 1)$. Fix $k \geq 1$ and $\delta \in (0, 1)$, if $l$ is sufficiently large, then with probability at least $1 - \delta$,*

$$\forall i, j \in [N], \mathcal{A}(Wx_i) \cap \mathcal{A}(Wx_j) = \emptyset.$$

*Proof.* Notice that the scale of $W$ and $x_1, x_2, \cdots, x_N$ do not affect either $\langle x_i, x_j \rangle / (\|x_i\|_2 \|x_j\|_2)$ or the activation pattern. Thus, we can assume $\|x_1\|_2 = \|x_2\|_2 = \cdots = \|x_N\|_2 = 1$ and each entry of $W$ is an i.i.d. standard Gaussian random variable.

Let $\beta = \frac{2 + \alpha + \sqrt{2 - \alpha^2}}{4}$ and $\varepsilon_\alpha$ be the same as mentioned in Lemma 9. Set $\varepsilon$ and $\beta'$ as

$$\varepsilon = \frac{\frac{1}{\beta} - 1}{2}, \quad \beta' = (1 + \varepsilon)\beta.$$

Now, set

$$\eta = \frac{\delta/100}{100k \log(N/\delta) \cdot (1 + 2/\sqrt{2(1 - \beta')})^m},$$

and let $R$ satisfies

$$\Pr_{Z \sim \chi_m^2}[Z \geq (1 + \varepsilon)^2 R^2] = \frac{\delta/100}{l}.$$

According to Lemma 8, if $l$ is sufficiently large, then $R$ is sufficiently large such that

$$\Pr_{Z \sim \chi_m^2}[Z \geq (1 + \varepsilon)^2 R^2] / \Pr_{Z \sim \chi_m^2}[(1 + \varepsilon)^2 R^2 \geq Z \geq R^2] \leq \eta.$$

Notice that for $t \in [l]$, $\|W_t\|_2^2$ is a random variable with $\chi_m^2$ distribution. Thus, $\Pr[\|W_t\|_2 \geq (1 + \varepsilon)R] = \frac{\delta/100}{l}$. By taking union bound over all $t \in [l]$, with probability at least $1 - \delta/100$, $\forall t \in [l], \|W_t\|_2 \leq (1 + \varepsilon)R$. In the remaining of the proof, we will condition on that $\forall t \in [l], \|W_t\|_2 \leq (1 + \varepsilon)R$. Consider $i, j \in [N], t \in [l]$, if $W_t x_i > \beta' R$, then we have

$$\frac{W_t x_i}{\|W_t\|_2} > \frac{\beta' R}{(1 + \varepsilon)R} \geq \beta'/(1 + \varepsilon) = \beta.$$

Due to Lemma 9, we have

$$\frac{W_t x_j}{\|W_t\|_2} < (1 - \varepsilon_\alpha)\beta.$$

Thus,

$$W_t x_j < (1 - \varepsilon_\alpha)\beta\|W_t\|_2 \leq (1 - \varepsilon_\alpha)\beta(1 + \varepsilon)R \leq (1 - \varepsilon_\alpha)\beta' R. \tag{20}$$

Notice that for $i \in [N], t \in [l]$, we have

$$\Pr[W_t x_i > \beta' R] \geq \Pr[\|W_t\|_2 \geq R] \Pr\left[\frac{W_t x_i}{\|W_t\|_2} \geq \beta'\right]$$

$$\geq \frac{\delta/100}{l} \cdot \frac{1}{\eta} \cdot \frac{1}{(1 + 1/\sqrt{2(1 - \beta')})^m}$$

$$\geq \frac{1}{l} \cdot 100k \log(N/\delta).$$

By Chernoff bound, with probability at least $1 - \delta/(100N)$,

$$\sum_{t=1}^{l} \mathbf{1}(W_t x_i > \beta' R) \geq k.$$

By taking union bound over $i \in [N]$, with probability at least $1 - \delta/100, \forall i \in [N]$,

$$\sum_{t=1}^{l} \mathbf{1}(W_t x_i > \beta' R) \geq k.$$

This implies that $\forall i \in [N]$, if $t \in \mathcal{A}(W x_i)$, then $W_t x_i > \beta' R$. Due to Equation (20), $\forall j \in [N]$, we have $W_t x_j < \beta' R$ which implies that $t \notin \mathcal{A}(W x_j)$. Thus, with probability at least $1 - \delta/50 \geq 1 - \delta$ probability, $\forall i \neq j, \mathcal{A}(W x_i) \cap \mathcal{A}(W x_j) = \emptyset$. $\qquad \square$

**Remark 1.** *Consider any $x_1, x_2, \cdots, x_N \in \mathbb{R}^m$ with $\|x_1\|_2 = \|x_2\|_2 = \cdots = \|x_N\|_2 = 1$. If $\forall i \neq j \in [N], \langle x_i, x_j \rangle \leq \alpha$ for some $\alpha \in (0.5, 1)$, then $|N| \leq (1 + 2/\sqrt{2(1 - \alpha)})^m$.*

*Proof.* Since $\langle x_i, x_j \rangle \leq \alpha, \|x_i - x_j\|_2^2 = \|x_i\|_2^2 + \|x_j\|_2^2 - 2\langle x_i, x_j \rangle \geq 2 - 2\alpha$. Let $\mathcal{S}$ be the unit sphere, i.e., $\mathcal{S} = \{x \in \mathbb{R}^m \mid \|x\|_2 = 1\}$. Due to Lemma 6, there is a $(\sqrt{2(1 - \alpha)}/2)$-net $\mathcal{N}$ of $\mathcal{S}$ with size at most $|\mathcal{N}| \leq (1 + 2/\sqrt{2(1 - \alpha)})^m$. Consider $x_i, x_j$, and $y \in \mathcal{N}$. By triangle inequality, if $\|x_i - y\|_2 < \sqrt{2(1 - \alpha)}/2$, then $\|x_j - y\|_2 > \sqrt{2(1 - \alpha)}/2$ due to $\|x_i - x_j\|_2 \geq \sqrt{2(1 - \alpha)}$. Since $\mathcal{N}$ is a net of $\mathcal{S}$, for each $x_i$, we can find a $y \in \mathcal{N}$ such that $\|x_i - y\|_2 < \sqrt{2(1 - \alpha)}/2$. Thus, we can conclude $N \leq |\mathcal{N}| \leq (1 + 2/\sqrt{2(1 - \alpha)})^m$. $\qquad \square$

**Theorem 11.** *Consider $N$ data points $x_1, x_2, \cdots, x_N \in \mathbb{R}^m$ with their corresponding labels $z_1, z_2, \cdots, z_N \in \mathbb{R}$ and a weight matrix $W \in \mathbb{R}^{l \times m}$ where each entry of $W$ is an i.i.d. sample drawn from the Gaussian distribution $N(0, 1/l)$. Suppose $\forall i \neq j \in [N], \langle x_i, x_j \rangle/(\|x_i\|_2 \|x_j\|_2) \leq \alpha$ for some $\alpha \in (0.5, 1)$. Fix $k \geq 1$ and $\delta \in (0, 1)$, if $l$ is sufficiently large, then with probability at least $1 - \delta$, there exists a vector $v \in \mathbb{R}^l$ such that*

$$\forall i \in [N], \langle v, \phi_k(W x_i) \rangle = z_i.$$

*Proof.* Due to Theorem 10, with probability at least $1 - \delta, \forall i \neq j, \mathcal{A}(W x_i) \cap \mathcal{A}(W x_j) = \emptyset$. Let $t_1, t_2, \cdots, t_N \in [l]$ such that $t_i \in \mathcal{A}(W x_i)$. Then $t_i \notin \mathcal{A}(W x_j)$ for $j \neq i$.
For each entry $v_t$, if $t = t_i$ for some $i \in [N]$, then set $v_t = z_i/(W_t x_i)$. Then for $i \in [N]$, we have

$$\langle v, \phi_k(W x_i) \rangle = \sum_{t \in \mathcal{A}(W x_i)} v_t \cdot W_t x_i = z_i/(W_{t_i} x_i) \cdot W_{t_i} x_i = z_i.$$

$\qquad \square$

# D  ADDITIONAL EXPERIMENTAL RESULTS

This section presents details of our experiment settings and additional results for evaluating and empirically understanding the robustness of $k$-WTA networks.

## D.1  EXPERIMENT SETTINGS

First, we describe the details of setting up the experiments described in Sec. 4. To compare $k$-WTA networks with their ReLU counterparts, we replace all ReLU activations in a network with $k$-WTA activation, while retaining all other modules (such as BatchNorm, Convolution, and pooling). To test on different network architectures, including ResNet18, DenseNet121, and Wide ResNet, we use the standard implementations that are publicly available[3]. All experiments are conducted using PyTorch framework.

**Training setups.**  We follow the same training procedure on CIFAR-10 and SVHN datasets. All the ReLU networks are trained with stochastic gradient descent (SGD) method with momentum=0.9. We use a learning rate 0.1 from the first to 50-th epoch and 0.01 from 50-th to 80-th epoch. To compare with ReLU networks, the k-WTA networks are trained in the same way as ReLU networks. All networks are trained with a batch size of 256.

For $k$-WTA networks with a sparsity ratio $\gamma = 0.1$, when adversarial training is not used, we train them incrementally (recall in Sec. 2.2). starting with $\gamma = 0.2$ for 50 epochs with SGD (using learning rate=0.1, momentum=0.9) and then decreasing $\gamma$ by 0.005 every 2 epochs until $\gamma$ reaches $0.1$.

When adversarial training is enabled, we use untargeted PGD attack with 8 iterations to construct adversarial examples. To train networks with TRADES (Zhang et al., 2019), we use the implementation of its original paper[4] with the parameter $1/\lambda = 6$, a value that reportedly leads to the best robustness according to the paper. To train networks with the free adversarial training method (Shafahi et al., 2019b), we implement the training algorithm by following the original paper. We set the parameter $m = 8$ as suggested in the paper.

**Attack setups.**  All attacks are evaluated under the $\ell_\infty$ metric, with perturbation size $\epsilon = 0.031$ (CIFAR-10) and 0.047 (SVHN) for pixels ranging in $[0, 1]$. We use Foolbox (Rauber et al., 2017), a third-party toolbox for evaluating adversarial robustness.

We use the following setups for generating adversarial examples in various attack methods: For PGD attack, we use 40 iterations with random start, the step size is 0.003. For C&W attack, we set the binary search step to 5, maximum number of iterations to 20, learning rate to 0.01, and initial constant to 0.01. For Deepfool, we use 20 steps and 10 sub-samples in its configuration. For momentum attack, we set the step size to 0.003 and number of iterations to 20. All other parameters are set by Foolbox to be its default values.

## D.2  EFFICACY OF INCREMENTAL TRAINING

We now report additional experiments to demonstrate the efficacy of the incremental fine-tuning method (described in Sec. 2.2). As shown in Figure 6 and described its caption, models trained with incremental fine-tuning (denoted as w/ FT in the plots' legend) performs better in terms of both standard accuracy (denoted as std in the plots' legend) and robust accuracy (denoted as Rob in the plots) when the $k$-WTA sparsity $\gamma < 0.2$, suggesting that fine-tuning is worthwhile when $\gamma$ is small.

## D.3  ADDITIONAL RESULTS ON CIFAR-10

**Tests on different network architectures.**  We evaluate the robustness of $k$-WTA on different network architectures, including ResNet-18, DenseNet-121 and WideResNet-22-10. The results are reported in Table 2, where similar to the notation used in Table 1 of the main text, $A_{rob}$ is calculated as the worst-case robustness, i.e., under the most effective attack among PGD, C&W, Deepfool and MIM. The training and attacking settings are same as other experiments described Sec. D.1.

As shown in Table 2, while the standard and robustness accuracies, $A_{std}$ and $A_{rob}$, vary on different network architectures, $k$-WTA networks consistently improves the worst-case robustness $A_{rob}$ over ReLU networks, no matter what the network architecture and training method are used.

---

[3] https://github.com/kuangliu/pytorch-cifar
[4] https://github.com/yaodongyu/TRADES

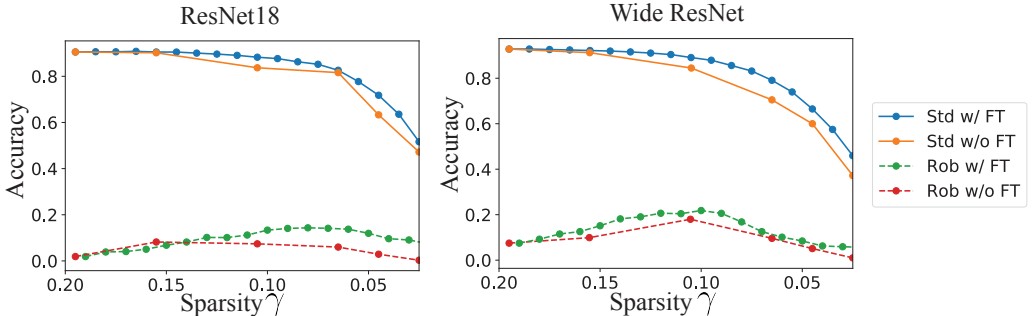

Figure 6: **Efficacy of incremental training.** We sweep through a range of sparsity ratios, and evaluate the standard and robust accuracies of two network structures (left: ResNet18 and right: Wide ResNet). We compare the performance differences between the regular training (i.e., training without incremental fine-tuning) and the training with incremental fine-tuning.

Table 2: Additional CIFAR-10 results.

| Training | Model | Activation | $A_{std}$ | $A_{rob}$ |
|---|---|---|---|---|
| Natural | ResNet-18 | ReLU | 92.9% | 0.0% |
| | | LWTA-0.1 | 82.8% | 3.7% |
| | | LWTA-0.2 | 84.6% | 0.9% |
| | | $k$-WTA-0.1 | 89.3% | **13.1%** |
| | | $k$-WTA-0.2 | 91.7% | 4.2% |
| AT | ResNet-18 | ReLU | 83.5% | 43.6% |
| | | LWTA-0.1 | 71.4% | 46.6% |
| | | LWTA-0.2 | 78.7% | 43.1% |
| | | $k$-WTA-0.1 | 78.9% | **50.7%** |
| | | $k$-WTA-0.2 | 81.4% | 47.4% |
| Natural | DenseNet-121 | ReLU | 93.6% | 0.0% |
| | | LWTA-0.1 | 86.1% | 4.6% |
| | | LWTA-0.2 | 88.5% | 1.4% |
| | | $k$-WTA-0.1 | 90.5% | **12.3%** |
| | | $k$-WTA-0.2 | 93.3% | 6.2% |
| AT | DenseNet-121 | ReLU | 84.2% | 46.3% |
| | | LWTA-0.1 | 74.0% | 49.1% |
| | | LWTA-0.2 | 80.2% | 44.9% |
| | | $k$-WTA-0.1 | 81.6% | **52.4%** |
| | | $k$-WTA-0.2 | 83.4% | 49.6% |
| Natural | WideResNet-22-10 | ReLU | 93.4% | 0.0% |
| | | LWTA-0.1 | 83.7% | 4.2% |
| | | LWTA-0.2 | 86.1% | 2.8% |
| | | $k$-WTA-0.1 | 88.6% | **18.3%** |
| | | $k$-WTA-0.2 | 92.7% | 7.4% |
| AT | WideResNet-22-10 | ReLU | 83.3% | 43.1% |
| | | LWTA-0.1 | 74.2% | 47.5% |
| | | LWTA-0.2 | 79.8% | 44.7% |
| | | $k$-WTA-0.1 | 78.9% | **50.4%** |
| | | $k$-WTA-0.2 | 82.4% | 47.1% |

**Comparison with LWTA.** We in addition compare $k$-WTA to LWTA activation (Srivastava et al., 2013; 2014). For fair comparisons, we use the same sparsity ratio $\gamma$ in both $k$-WTA and LWTA. As shown in Table 2, on all network architectures and training methods we tested, $k$-WTA networks consistently have better robustness performance than LWTA networks (in terms of both $A_{std}$ and $A_{rob}$). These results suggest that $k$-WTA is more suitable then LWTA for defending against adversarial attacks.

**Transfer attack.** Since a $k$-WTA network is architecturally similar to its ReLU counterpart—with the only difference being the activation—we evaluate their robustness under (black-box) transfer

Table 3: Transferability test on CIFAR-10.

| Target Model | Source Model | | | |
|---|---|---|---|---|
| | ReLU | $k$-WTA-0.1 | ReLU (AT) | $k$-WTA-0.1 (AT) |
| ReLU | 4.8% | 75.5% | 59.4% | 84.7% |
| $k$-WTA-0.1 | 61.2% | 71.2% | 67.8% | 86.4% |
| ReLU (AT) | 62.7% | 80.9% | 61.6% | 78.6% |
| $k$-WTA-0.1 (AT) | 79.2% | 78.6% | 69.2% | 67.2% |

Table 4: White-box attack results on MNIST dataset.

| Activation | Training | $A_{std}$ | $A_{rob}$ | Activation | Training | $A_{std}$ | $A_{rob}$ |
|---|---|---|---|---|---|---|---|
| ReLU | Natural | 99.4% | 0.0% | $k$-WTA-0.1 | Natural | 99.3% | **62.2%** |
| | AT | 99.2% | 95.0% | | AT | 99.2% | **96.4%** |
| | TRADES | 99.2% | 96.0% | | TRADES | 99.0% | **96.9%** |
| | FAT | 98.2% | 94.7% | | FAT | 98.1% | **96.0%** |

attacks across $k$-WTA and ReLU networks. To this end, we build a ReLU and a $k$-WTA-0.1 network on ResNet-18, and train both networks with natural (non-adversarial) training as well as adversarial training. This gives us four different models denoted (in Table 3) as ReLU, $k$-WTA-0.1, ReLU (AT), and $k$-WTA-0.1 (AT). We then launch transfer attacks across each pair of models. We also consider by-far the strongest black-box attack (according to Papernot et al. (2017)): for the same model, for example, a $k$-WTA-0.1 network optimized by adversarial training, we train two independent versions, each with a different random initialization, and apply the transfer attacks across the two versions.

The results are reported in Table 3, where each row corresponds to a target (attacked) model, and each column corresponds to a source model from which the adversarial examples are generated. On the diagonal line of Table 3, each entry corresponds to the robustness under aforementioned transfer attacks across the two versions of the same models.

The results suggest that **1)** it is more difficult to transfer attack $k$-WTA networks than ReLU networks using adversarial examples from other models, and **2)** it is also more difficult to use adversarial examples of a $k$-WTA network to attack other models. In a sense, the adversarial examples of a $k$-WTA network tend to be "disjoint" from the adversarial examples of a ReLU network, despite their architectural similarity. Inspecting the diagonal entries of Table 3, we also find that $k$-WTA networks are more robust than their ReLU counterparts under the strongest black-box attack (Papernot et al., 2017) (i.e., transfer attacks across two different versions of the same model).

### D.4 MNIST RESULTS

On MNIST dataset, we conduct experiments with an adversarial perturbation size $\epsilon$=0.3 for pixels ranging in $[0, 1]$. We use Stochastic Gradient Descent (SGD) with learning rate=0.01 and momentum=0.9 to train a 3-layer CNN. The training takes 20 epochs for all the methods we evaluate. The robust accuracy are evaluated under PGD attacks that take 20 iterations with random initialization and a step size of 0.03.

The results are summarized in Table 4. Again, $k$-WTA activation consistently improves robustness under all different training methods. Even with natural (non-adversarial) training, the resulting $k$-WTA network still has 62.2% robust accuracy, significantly outperforming ReLU network.

### D.5 ROBUSTNESS WITH RESPECT TO NATURAL PERTURBATIONS

We also evaluate the robustness of $k$-WTA networks under (non-adversarial) natural perturbations. We evaluated various types of perturbations, including adding Gaussian noise to the input image (std=0.05/0.1), random translation (maximum 5 pixel), random rotation (maximum 10 degrees) and color jittering (i.e., randomly changing the brightness, contrast and saturation of an image, with a maximum perturbation of 0.4), following Hendrycks & Dietterich (2019). The results are summerized in Table 5, in which all models are ResNet18.

We found that under all the tested perturbations, the accuracy drops in $k$-WTA networks are *no* worse than those in ReLU networks. Note that in these tests, all our models are trained with standard data

Table 5: Robustness under natural perturbations.

| Model | Clean | GN(0.05) | GN (0.1) | Translation | Rotation | ColorJitter |
|-------|-------|----------|----------|-------------|----------|-------------|
| ReLU | 92.9% | 69.7% | 27.0% | 92.3% | 88.9% | 90.7% |
| $k$-WTA-0.1 | 89.3% | 80.2% | 50.9% | 89.1% | 86.8% | 87.4% |
| $k$-WTA-0.2 | 91.7% | 77.9% | 39.6% | 91.2% | 88.9% | 89.4% |
| ReLU (AT) | 83.5% | 80.3% | 73.9% | 80.5% | 79.8% | 74.9% |
| $k$-WTA-0.1 (AT) | 78.9% | 77.8% | 69.8% | 75.9% | 74.7% | 68.7% |
| $k$-WTA-0.2 (AT) | 81.4% | 80.6% | 70.9% | 79.8% | 78.6% | 74.7% |

augmentations (e.g., random crop and random flip); they are not specifically trained to avert the tested perturbations.

We also highlight an interesting finding here. Adding Gaussian noise leads to a large accuracy drop (e.g., from 92.9% to 69.7%/27.0% as shown in the table) on naturally trained ReLU network, but in $k$-WTA networks (especially $k$-WTA-0.1), the corresponding accuracy drop is much smaller (e.g., from 89.3% to 80.2%/50.9%). We conjecture that the dense discontinuities in $k$-WTA networks (recall Figure 5) effectively add noise to the input distribution, thus making the model more robust against input noise.

### D.6 LOSS LANDSCAPE VISUALIZATION

In addition to the experiments shown in Figure 5 and Sec. 4.3 of the main text, we further we visualize the loss landscapes of $k$-WTA networks when different sparsity ratios $\gamma$ are used. The plots are shown in Figure 7, produced in the same way as Figure 5 described in Sec. 4.3.

As analyzed in Sec. 3, a larger $\gamma$ tends to smooth the loss surface of the $k$-WTA network with respect to the input, while a smaller $\gamma$ renders the loss surface more discontinuous and "spiky". In addition, adversarial training tends to reduce the range of the loss values—a similar phenomenon in ReLU networks has already been reported Madry et al. (2017); Tramèr et al. (2017)—but that does not mean that the loss surface becomes smoother; the loss surface remains spiky.

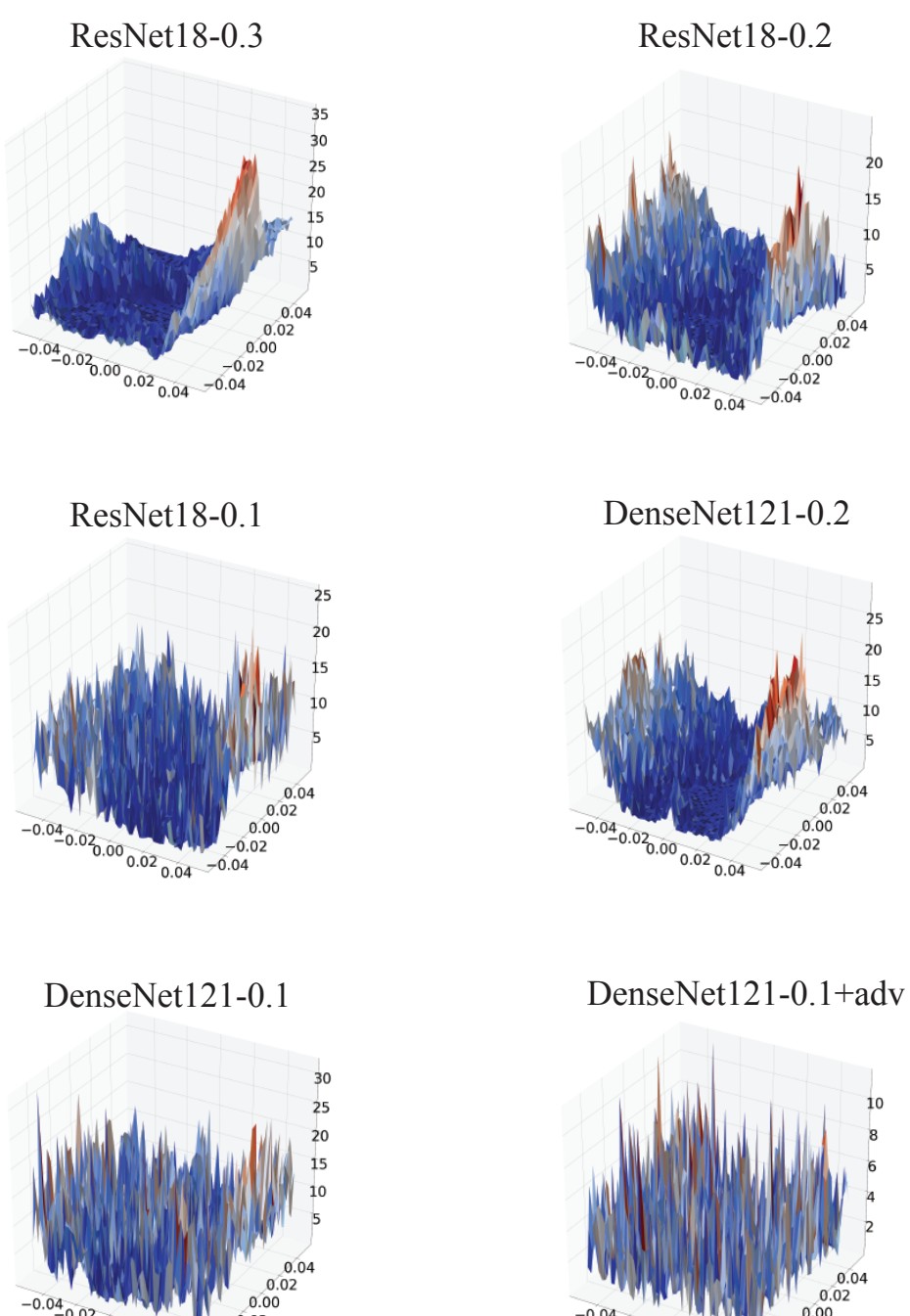

Figure 7: Visualization of ResNet18 and DenseNet121 with different $\gamma$ values. The last one (DenseNet121-0.1+adv) is the result using adversarial training. The others are optimized using natural (non-adversarial) training.

