# OpenReview forum: "Enhancing Adversarial Defense by k-Winners-Take-All"
_ICLR.cc/2020/Conference — Accept (Spotlight)_

### Official Review · AnonReviewer1 · 2019-10-21
**Official Blind Review #1**

**Rating:** 8

**Review:**

This paper addresses the important question of improving the robustness of deep neural networks against adversarial attacks. The authors propose a surprisingly simple measure to improve adversarial robustness, namely replacing typical activation functions such as ReLU with a k-winners-take-all (k-WTA) functions, whereby the k largest values of the input vector are copied, and all other elements of the output vector are set to zero. Since the size of input and output maps varies drastically within networks, the authors instead use a sparsity ratio \gamma that calculates k as a function of input size. k-WTA networks can be trained without special treatment, but for low \gamma values the authors propose a training schedule, whereby \gamma is slowly reduced, then re-training takes place, until the desired value of \gamma is reached. The presented effect is backed up by extensive theoretical investigations that relate the increased robustness to the dense introduction of discontinuities, which makes gradient-based adversarial attacks harder. A small change in an input signal can change the identity of the "winning" inputs, and thus in a sub-sequent matrix multiplication make use of other rows or columns, thus allowing arbitrarily large effects due to small input variations. Empirical evaluations in CIFAR and SVHN for a variety of attacks and defense mechanisms demonstrate the desired effects, and illustrate the loss landscapes due to using k-WTA.

I think the paper is a valuable and novel contribution to an important topic, and is definitely suitable for publication at ICLR. In principle there is just one novel idea, namely using k-WTA activations to improve adversarial robustness, but this claim is investigated thoroughly, in theory, and demonstrated convincingly in experiments. The paper is well written and tries to address all potential questions one might have surrounding the basic idea. There is code available, and the idea should be simple to implement in practice, so I would expect this paper to have large impact on the study of adversarial robustness.

I appreciate the thorough proofs of the claims in section C of the appendix, but I did not review all proofs in  detail.

Potential weaknesses that should be addressed:
1. Section 4: I would propose to fully define A_rob here or at least provide a reference.
2. From Table 1 it seems that using k-WTA leads to a quite noticeable drop in standard accuracy, especially for sparse \gamma, which leads to the best adversarial robustness. Can you please comment on whether the full ReLU accuracy in the natural case can always be recovered by k-WTA networks, e.g. with larger \gamma?
3. Since small changes have a big effect in k-WTA, it should be investigated how robust the k-WTA networks are with respect to more natural perturbations, e.g. noisy input, blurring, translations, rotations, occlusions, etc. as introduced in (Hendrycks and Dietterich, 2019). It would be critical if such perturbations have a stronger effect on k-WTA.
4. Is there any intuition about whether k-WTA should be used everywhere in a deep network, or whether it makes sense to mix k-WTA and ReLU functions?

Minor comments:
5. WTA networks are very popular in computational neuroscience and are even hypothesized to represent canonical microcircuit functions (see e.g. Douglas, Martin, Whitteridge, 1989, Neural Computation, and many follow-up articles). You cite the work of Maass, 2000a,b already, it could be interesting to link your work to other papers in that field who motivate WTA from a biological perspective.

**Experience Assessment:**

I have read many papers in this area.

**Review Assessment: Checking Correctness Of Derivations And Theory:**

I did not assess the derivations or theory.

**Review Assessment: Checking Correctness Of Experiments:**

I assessed the sensibility of the experiments.

**Review Assessment: Thoroughness In Paper Reading:**

I read the paper at least twice and used my best judgement in assessing the paper.

---

> ### Author Response · Authors · 2019-11-08
> **AnonReviewer1 Response**
>
> We thank you for your effort in reviewing our paper and providing constructive feedback. We respond to your main points below.
>
> > “Section 4: I would propose to fully define A_rob here or at least provide a reference.”
>
> We follow the definition of $A_{rob}$ used in other papers, namely, (number of correctly recognized adversarial image) divided by (number of all testing images). We will make this clear in our revision.
>
> > “From Table 1 it seems that using k-WTA leads to a quite noticeable drop in standard accuracy, ...”
>
> We found from our experiments that as $\gamma$ increases, the network tends to yield higher standard accuracy but lower robust accuracy. When $\gamma$ is relatively large (i.e., $\gamma=0.3$), the $k$-WTA network’s standard accuracy is comparable to ReLU network, but its robust accuracy is not significantly better than ReLU network. For example, on CIFAR-10 dataset, we have the following accuracy results.
>
> ==========================================
> Model                     A_std       A_rob(under PGD-20)
> ———————————————————————
> ReLU                      92.9%       0.0%
> kWTA-0.3               92.7%       1.6%
> ———————————————————————
> ReLU+AT               83.5%        46.3%
> kWTA-0.3+AT        83.8%        46.9%
> ==========================================
>
> The is because when $\gamma$ is large (e.g., >0.3), the $k$-WTA network becomes less discontinuous. This trend can be seen in Figure 7 (in the supplementary document): the loss landscape is much smoother for $k$-WTA-0.3 (wherein $\gamma=0.3$) in comparison to smaller $\gamma$ values. Indeed, Srivastava et al. (2013) also showed that WTA-type activations (not precisely our $k$-WTA though) can offer an accuracy comparable to ReLU. We believe $k$-WTA can always achieve the standard accuracy comparable to ReLU, as long as $\gamma$ is sufficiently large. But then, the robust accuracy may not be as high. In this sense, $\gamma$ can be viewed as a knob that controls the tradeoff between standard and robust accuracies.
>
> > “Since small changes have a big effect in k-WTA, it should be investigated how robust the k-WTA networks are with respect to more natural perturbations ...”
>
> Thanks for your suggested test. We tested various types of perturbations, including adding Gaussian noise to the input image (std=0.05/0.1), random translation (maximum 5 pixel), random rotation (maximum 10 degrees) and color jittering (i.e., randomly changing the brightness, contrast and saturation of an image, with a maximum perturbation of 0.4), all following [Hendrycks and Dietterich 2019]. The resulting accuracies are summarized in the following table.
> ========================================================================
> Model                         Clean     GN(0.05)   GN(0.1)     Translation    Rotation    ColorJitter
>
> ReLU                           92.9%      69.7%        27.0%        92.3%              88.9%        90.7%
> kWTA-0.1                    89.3%      80.2%        50.9%        89.1%              86.8%        87.4%
> kWTA-0.2                    91.7%      77.9%        39.6%        91.2%              88.9%        89.4%
>
> ReLU+AT                     83.5%      80.3%        73.9%       80.5%               79.8%       74.9%
> kWTA-0.1+AT              78.9%      77.8%       69.8%        75.9%               74.7%       68.7%
> kWTA-0.2+AT              81.4%      80.6%       70.9%        79.8%               78.6%       74.7%
> ========================================================================
>
> We found that under all the tested perturbations, the accuracy drops in $k$-WTA networks are *no* worse than those in ReLU networks. We would like to stress that in these tests, all our models are trained with standard data augmentations (e.g., random crop and random flip); they are not specifically trained to avert the tested perturbations.
>
> We would also like to highlight an interesting finding here. Adding Gaussian noise leads to a large accuracy drop (e.g., from 92.9% to 69.7% as shown in the table) on naturally trained ReLU network, but in $k$-WTA networks (especially $k$-WTA-0.1), the corresponding accuracy drop is much smaller (e.g., from 89.3% to 80.2%). We conjecture that the dense discontinuities in $k$-WTA networks (recall Figure 7) effectively add noise to the input distribution, thus making the model more robust against input noise.
>
> > “Is there any intuition about whether k-WTA should be used everywhere in a deep network ...”
>
> Our intuition is that $k$-WTA can improve the robustness while ReLU is easier for training. Therefore, $k$-WTA offers a simple way to make a compromise, that is, slightly sacrificing standard accuracy in exchange for robust accuracy. We believe that such a tradeoff between standard and robust accuracies will be affected if one chooses to mix the two types of activation functions in a network. Better understanding of this tradeoff for a mixture of activation functions is certainly an interesting direction for future study.

---

> > ### Author Response · Authors · 2019-11-08
> > **Minor Comment Response**
> >
> > > “Minor comments: WTA networks are very popular in computational neuroscience...”
> >
> > Thanks for the comments. The connection between $k$-WTA and those in computational neuroscience is indeed interesting and intriguing. We will discuss this connection in our paper revision.

---

### Official Review · AnonReviewer2 · 2019-10-21
**Official Blind Review #2**

**Rating:** 8

**Review:**

The authors propose using k-winner take all (k-WTA) activation functions to prevent white box adversarial attacks. A k-WTA activation functions outputs the k highest activations in a layer while setting all other activations to zero. The reasoning given by the authors is that k-WTA activation functions have many discontinuities with respect to the input space. This makes it more difficult for attacks to use gradient information. The authors note that networks with k-WTA activation functions are still easy to train because, for a given input, the sub-network that is activated becomes more stable as training progresses. Therefore, it is not as discontinuous in the parameter space.

The authors test their method with 5 different adversarial attacks and train with 4 different training methods. They use the CIFAR10 and SVNH datasets.

The experiments showed that using k-WTA activation functions resulted in consistent improvements over ReLU activation functions in model robustness to white-box adversarial attacks when training with and without adversarial training methods. While, in the worst case, ReLU networks were around 50%-58% accurate in the face of adversarial attacks, k-WTA has accuracy that is usually 5%-17% higher.

While the novelty of this paper is low, the switch from ReLU to k-WTA appear relatively simple and yields better results than that of ReLU.

Other comments:
I don't think that this claim can be made without experimental evidence and should be removed:
"We are not aware of any possible smooth approximation of a k-WTA network to launch BPDA attacks.
Even if hypothetically there exists a smooth approximation of k-WTA activation, that approximation
has to be applied to every layer. Then the network would accumulate the approximation error at each
layer rapidly so that any gradient-estimation-based attack (such as BPDA) will be defeated."

Question:
I see the \gamma parameter of k-WTA is updated with a certain schedule that includes some finetuning. Including this finetuning, is the final k-WTA network trained with the same number of iterations as the ReLU network? Are all the other hyperparameters the same?

** After Author Response **
Changing from weak accept to accept

The authors have addressed my concerns and I believe the paper can provide significant value to those interested in adversarial robustness.

**Experience Assessment:**

I have read many papers in this area.

**Review Assessment: Checking Correctness Of Derivations And Theory:**

I assessed the sensibility of the derivations and theory.

**Review Assessment: Checking Correctness Of Experiments:**

I assessed the sensibility of the experiments.

**Review Assessment: Thoroughness In Paper Reading:**

I read the paper at least twice and used my best judgement in assessing the paper.

---

> ### Author Response · Authors · 2019-11-07
> **AnonReviewer2 Response**
>
> We thank you for your effort in reviewing our paper and providing insightful feedback. We respond to your main points below.
>
> > “While the novelty of this paper is low, ...”
>
> While we understand that novelty is perhaps subjective to individual reviewers, here we would like to stress the novelty that we wish to deliver in this paper: namely, 1) the simplicity and efficacy of the use of $k$-WTA for improving adversarial robustness and 2) the theoretical insights that we provide to understand why $k$-WTA helps to improve the robustness. It is the simplicity that makes our method easy to adopt in nearly all existing networks; it is the efficacy for improving robustness in a wide range of setups that makes our method worth adopting in practice; and it is the theoretical insights---the analysis looking into the simple idea---that sheds some light toward better understanding of $k$-WTA in particular and adversarial robustness in general.
>
> > “I don't think that this claim can be made without experimental evidence and should be removed ...”
>
> We propose to remove the sentences starting from “Even if hypothetically there exists ...” as you suggested. The first sentence (“We are not aware of ...”) is meant to state the fact that to our knowledge, how to smoothly approximate the $k$-WTA network remains open. The point we wish to convey is that we are not aware of any approach to launch BPDA attacks.
>
> > “Question:”
>
> Yes, our $k$-WTA network is trained with the same number of iterations as the ReLU network, and all other hyperparameters are the same. We will clarify these points in the paper.

---

### Official Review · AnonReviewer3 · 2019-10-27
**Official Blind Review #3**

**Rating:** 8

**Review:**

This paper suggests using the activation function k-Winners-Take-All (k-WTA) in deep neural networks to enhance the performance of adversarial defense. Their experiments show that the robustness is improved when they simply change the activation function to k-WTA. They also give reasonable theoretical analysis for their approach.

I find that the idea is simple and elegant. Since they only change the activation function, their approach can be easily applied to almost all network structures and training processes. Their experiments show that the robust accuracies are significantly improved on all evaluated methods when they use the k-WTA activation function. I also appreciate their theoretical analysis. They show that k-WTA makes the network very discontinuous with respect to the input x, and thus the adversary could not get useful gradient information. In contrast, if the network is wide enough, then the discontinuities with respect to the weights w is sparse. This is why the network is still trainable though itself is not continuous.

The paper is also well-written and easy to follow. I recommend the acceptance of the paper.

One limitation of this paper is that their approach mainly focuses on defending gradient based attack. But I agree that the gradient based attack is currently almost the best attack method.

A minor question:
- How do we choose k in general? What is the behaviour for different k?


**Experience Assessment:**

I have published one or two papers in this area.

**Review Assessment: Checking Correctness Of Derivations And Theory:**

I carefully checked the derivations and theory.

**Review Assessment: Checking Correctness Of Experiments:**

I carefully checked the experiments.

**Review Assessment: Thoroughness In Paper Reading:**

I read the paper at least twice and used my best judgement in assessing the paper.

---

> ### Author Response · Authors · 2019-11-09
> **AnonReviewer3 Response**
>
> We thank you for your comments and observations.
>
> Instead of directly specifying $k$, we use a parameter $\gamma \in (0, 1)$ called sparsity ratio. If a layer has an output dimension $N$, then its $k$-WTA activation has $k = \gamma/N$. From our experiments, we found that $\gamma=0.1$ usually yields the best performance in terms of robust accuracy. If we increase $\gamma$, we will get a better standard accuracy but lower robust accuracy. We also found that when $\gamma\geq 0.3$, $k$-WTA networks can reach similar standard accuracy as ReLU networks, but its robust accuracy becomes lower.

---

### Author Response · Authors · 2019-11-14
**Revision posted**


Dear reviewers,

We have posted a revision of our paper. Beside fixing some typos and format issues, our major changes include:
* an experiment testing $k$-WTA networks against various types of (non-adversarial) perturbations.
* clarification of $A_{rob}$
* pointing out the connection of $k$-WTA to computational neuroscience
* clarifying the statement in related work section

Please kindly let us know if you have any further comments. Thanks again for your effort in reviewing our paper!

---

### Decision · Program_Chairs · 2019-12-19

**Decision:**

Accept (Spotlight)

**Comment:**

This paper presents new non-linearity function which specially affects regions of the model which are densely valued. The non-linearity is simple: it retains only top-k highest units from the input, while truncating the rest to zero. This also makes the models more robust to adversarial defense which depend on the gradients. The non-linearity function is shown to have better adversarial robustness on CIFAR-10 and SVHN datasets. The paper also presents theoretical analysis for why the non-linearity is a good function.

The authors have already incorporated major suggestions by the reviewers and the paper can make significant impact on the community. Thus, I recommend its acceptance.